# *PAX3-FOXO1* transgenic zebrafish models identify *HES3* as a mediator of rhabdomyosarcoma tumorigenesis

Genevieve C Kendall[1,2], Sarah Watson[3], Lin Xu[1], Collette A LaVigne[1,2], Whitney Murchison[1], Dinesh Rakheja[1,4], Stephen X Skapek[1], Franck Tirode[5], Olivier Delattre[3,6,7], James F Amatruda[1,2,8]*

[1]Department of Pediatrics, UT Southwestern Medical Center, Dallas, United States; [2]Department of Molecular Biology, UT Southwestern Medical Center, Dallas, United States; [3]Institut Curie, Paris Sciences et Lettres (PSL) Research University, Inserm U830, Institut Curie, Paris Sciences et Lettres (PSL) Research University, Paris, France; [4]Department of Pathology, UT Southwestern Medical Center, Dallas, United States; [5]Univ Lyon, Université Claude Bernard Lyon 1, INSERM 1052, CNRS 5286, Centre LéonBérard, Lyon, France; [6]INSERM U80, Institute Curie Research Center, Paris, France; [7]Institut Curie Hospital Group, Unité de Génétique Somatique, Paris, France; [8]Department of Internal Medicine, UT Southwestern Medical Center, Dallas, United States

*For correspondence:
james.amatruda@utsouthwestern.edu

**Abstract** Alveolar rhabdomyosarcoma is a pediatric soft-tissue sarcoma caused by *PAX3/7-FOXO1* fusion oncogenes and is characterized by impaired skeletal muscle development. We developed human *PAX3-FOXO1* -driven zebrafish models of tumorigenesis and found that *PAX3-FOXO1* exhibits discrete cell lineage susceptibility and transformation. Tumors developed by 1.6–19 months and were primitive neuroectodermal tumors or rhabdomyosarcoma. We applied this *PAX3-FOXO1* transgenic zebrafish model to study how *PAX3-FOXO1* leverages early developmental pathways for oncogenesis and found that *her3* is a unique target. Ectopic expression of the *her3* human ortholog, *HES3*, inhibits myogenesis in zebrafish and mammalian cells, recapitulating the arrested muscle development characteristic of rhabdomyosarcoma. In patients, *HES3* is overexpressed in fusion-positive versus fusion-negative tumors. Finally, *HES3* overexpression is associated with reduced survival in patients in the context of the fusion. Our novel zebrafish rhabdomyosarcoma model identifies a new *PAX3-FOXO1* target, *her3*/*HES3*, that contributes to impaired myogenic differentiation and has prognostic significance in human disease.
DOI: https://doi.org/10.7554/eLife.33800.001

## Introduction

Rhabdomyosarcoma (RMS) presents as a solid tumor that displays characteristics of primitive skeletal muscle (*Parham and Ellison, 2006*). There are two major histological subtypes, embryonal (ERMS) and alveolar rhabdomyosarcoma (ARMS), with ARMS being more aggressive and particularly prone to metastasis (*Williamson et al., 2010*; *Skapek et al., 2013*). At the genomic level, ERMS is characterized by frequent alterations affecting *RAS* signaling, although this still represents a minority of the cases (*Stratton et al., 1989*; *Langenau et al., 2007*; *Martinelli et al., 2009*). The defining oncogenic event in ARMS is a t(1;13) or t(2;13) chromosomal translocation in which the *PAX7* or *PAX3* DNA-binding domain, respectively, is fused to the *FOXO1* transactivation domain to create a *PAX3/7-FOXO1* chimeric oncogene (*Barr et al., 1993*; *Galili et al., 1993*; *Shapiro et al., 1993*; *Davis et al.,*

**eLife digest** One of the most common cancers in children and adolescents is rhabdomyosarcoma, a cancer of soft tissue such as muscle, tendon or cartilage. The fusion of DNA on two different chromosomes causes the most aggressive form of rhabdomyosarcoma. The fused DNA produces an abnormal protein called PAX3-FOXO1. During normal muscle development, a subset of rapidly growing cells eventually slow down and form mature, working muscle cells. It is still unclear how exactly rhabdomyosarcoma develops, but it is thought that PAX3-FOXO1 stops muscle cells from maturing and the cells that grow out of control result in a tumor. Learning how PAX3-FOXO1 hijacks normal muscle development could lead to new treatments for rhabdomyosarcoma.

One treatment approach is to slow the growth of a tumor and force the cells to mature. Then, young patients might avoid chemotherapy or radiation treatments and their side effects. Efforts to improve treatment for this type of cancer face several obstacles. Currently, only one vertebrate animal model of the disease is available to test drugs, and it is still not known how PAX3-FOXO1 causes healthy cells to become cancerous. It is also hard to turn off PAX3-FOXO1 itself, so scientists must find additional proteins that collaborate with it to target with drugs.

Now, Kendall et al. show that genetically engineered zebrafish with human PAX3-FOXO1 develop rhabdomyosarcoma-like tumors. Experiments on these zebrafish showed that the protein turns on a gene called her3. Humans have a similar gene called HES3. In zebrafish or mouse cells, human HES3 interferes with muscle-cell maturation and allows cells that acquire PAX3-FOXO1 to persist during development instead of dying. It also increases the cell growth and cancerous behavior in human tumor cells. Kendall et al. further looked at HES3 levels in tumors collected from patients with rhabdomyosarcoma and found that having higher levels of HES3 increased the risk of death from the cancer.

Human rhabdomyosarcoma tumors with high HES3 levels were also more likely to have certain cell-growth and cell-differentiation related proteins. Drugs that turn off or modify the activity of these proteins already exist. Testing these drugs that target processes such as cell growth in the zebrafish with rhabdomyosarcoma-like tumors may help scientists determine if they reduce tumor growth. If they do, additional trials could determine if they would help patients with rhabdomyosarcoma.

DOI: https://doi.org/10.7554/eLife.33800.002

1994). The PAX3-FOXO1 fusion is the most prevalent fusion in the disease, and functions as an aberrant transcription factor that is expressed in the nucleus and deregulates gene expression signatures (del Peso et al., 1999; Fredericks et al., 1995; Barber et al., 2002; Khan et al., 1999). This activity is the predominant cellular insult required for transformation.

The PAX3/7-FOXO1 oncogenes remain intractable to therapeutic targeting, impeding the development of effective precision medicine therapies. Fusions are notoriously difficult to model in animals, hence the limited availability of vertebrate animal models of this disease. Furthermore, there is a narrow understanding of the cellular origin of RMS, making it difficult to define the expression pattern required for tumorigenesis (Hettmer and Wagers, 2010). Zebrafish are a complementary model system that can address these genetic and cellular issues. Advantages of zebrafish systems are two-fold: (1) they provide insight into the underlying biology of how cancer genes behave in a complex environment and (2) provide a platform for translational drug discovery efforts. Such strengths are intrinsically important for translational models of pediatric disease.

Here, we describe human PAX3-FOXO1-driven zebrafish models. We implemented our zebrafish models to provide the appropriate context to understand the behavior of PAX3-FOXO1 during development and tumorigenesis. The tumor presentation spectrum identified three distinct cellular contexts that are susceptible to transformation, generating insight into basic mechanisms of PAX3-FOXO1 tumorigenesis and human rhabdomyosarcoma. By applying our zebrafish RMS model, we found a novel PAX3-FOXO1 target, her3/HES3. HES3 is a member of the HES family of basic helix-loop-helix transcription factors, which function as direct or indirect transcriptional activators or repressors, thus regulating gene expression and epigenetic identity (Kageyama et al., 2007). HES3 is expressed in the developing brain and inhibits differentiation of neural stem cells

(*Hatakeyama et al., 2004*). In cancer, *HES3* is expressed in glioblastoma cell culture, and co-localizes with additional markers of stemness in the mouse brain (*Park et al., 2013*; *Poser et al., 2013*; *Katoh and Katoh, 2007*). However, its role as a cooperating gene in *PAX3/7-FOXO1* fusion-positive rhabdomyosarcoma has never been described. Taken together, this model represents a novel strategy to identify new targets and biomarkers in the context of human disease and contributes to our understanding of RMS biology by defining the earliest tumor initiation events.

## Results

### A transgenic zebrafish model of human *PAX3-FOXO1* driven tumorigenesis

To develop a new vertebrate model of *PAX3-FOXO1*-dependent tumorigenesis, we performed a survey of promoters driving *PAX3-FOXO1* expression. These promoters represent ubiquitous (beta actin, CMV, ubiquitin), hematologic (fli1), muscle (unc503), neural crest (mitfa) expression, and a gene trap approach. Selected promoters were chosen because of their relevance in the disease as implicated lineages for the cell of origin or for their capacity to drive *PAX3-FOXO1* at high levels of expression. Further, all promoters had been previously validated as functional in zebrafish, with data from our group underscoring the beta actin promoter as a successful expression system for *EWS-FLI1* transgenic models of Ewing sarcoma (*Leacock et al., 2012*). Human *PAX3-FOXO1* was integrated into the zebrafish genome utilizing the Tol2 transposon-based system and microinjection in a stable mosaic manner. Genomic integration and transgene expression were tracked using a GFP or mCherry fluorescent protein linked to the coding sequence of *PAX3-FOXO1* with a viral 2A sequence (*Figure 1A–D*). This results in equimolar expression of both genes on the same mRNA transcript, yet translation as independent proteins. Zebrafish were monitored for up to 19 months. Using this strategy, we identified fusion-oncogene driven tumors first based on gross morphology and then by screening suspected tumors for fluorescent signal.

Human *PAX3-FOXO1* under the control of different promoters had unique survival and transforming properties suggesting cell-of-origin specificity to oncogenesis. Tested promoters had a negative impact on early survival (up to 30 days of life), but to different extents (*Figure 1—figure supplement 1*). Only a subset of the promoters that were tested induced tumor formation, including the beta actin (9% of injected zebrafish), CMV (16% of injected zebrafish), and ubiquitin (1.8% of injected zebrafish) promoters (*Supplementary file 1*). The *fli1*, *unc503*, *mitfa*, and gene trap approach driving *PAX3-FOXO1* were not transforming (*Supplementary file 2*).

The three promoters (beta actin, CMV, ubiquitin) that induced *PAX3-FOXO1* transformation in zebrafish had varied requirements for $tp53^{M214K}$ as a sensitizing mutation. The beta actin promoter driving the expression of *PAX3-FOXO1* is tumorigenic in zebrafish, both in a wildtype and a p53-deficient genetic background. In *tp53* wild-type zebrafish, BetaActin-*PAX3FOXO1* tumors began developing at 3 months of age in 5% of injected zebrafish in the head. Diagnosis of tumors was made by hematoxylin and eosin staining and evaluation by light microscopy. Based on this analysis, the majority of beta-actin-driven *PAX3-FOXO1* zebrafish tumors were consistent with primitive neuroectodermal tumors (*Figure 1B*). However, beta-actin-driven *PAX3-FOXO1* injected into the $tp53^{M214K/M214K}$ mutant resulted in one undifferentiated sarcoma after 412 days.

The CMV promoter restricting *PAX3-FOXO1* expression results in RMS tumors that present in the skeletal muscle of the back. Histological analysis of the zebrafish tumors was consistent with human RMS (*Figure 1C*). This was true only in the context of the $tp53^{M214K}$ mutation, indicating that the *tp53* mutation is sensitizing to and favors RMS. These findings are concordant with the *Pax3-Foxo1* RMS mouse model, in which *Tp53* deletion, or other secondary cooperating mutations are required for RMS development (*Keller et al., 2004*). Further, patients with Li Fraumeni syndrome develop RMS as part of the spectrum of the human disease (*Li and Fraumeni, 1969*).

The ubiquitin promoter driving *PAX3-FOXO1* generated one undifferentiated sarcoma by 378 days of age in a wild-type genetic background (*Figure 1D*). We performed RNA-seq on zebrafish tumors derived from all three promoters (beta actin, CMV, ubiquitin), and detected *PAX3* and *FOXO1* junction spanning reads indicating that human *PAX3-FOXO1* is expressed in fluorescent tumors (*Figure 1E*, *Figure 1—figure supplement 2*). These data demonstrate that the human

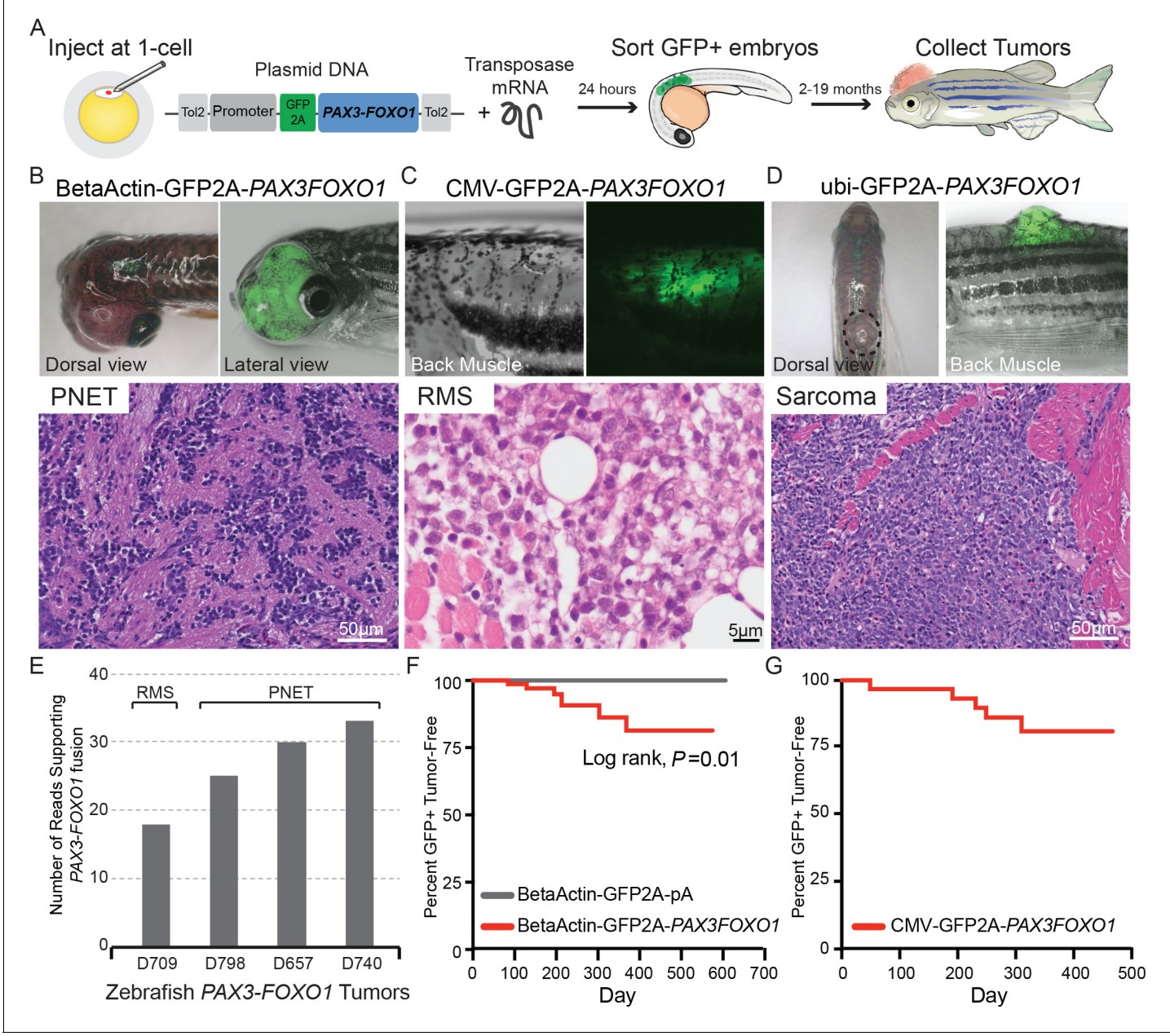

**Figure 1.** Zebrafish models of human *PAX3-FOXO1* tumorigenesis. (**A**) Zebrafish were injected at the single-cell stage with mosaic GFP2A-tagged human *PAX3-FOXO1* under the control of various promoters. At 24 hr old, embryos were sorted for GFP expression indicating successful injections (typically 99% GFP+) and were allowed to grow and monitored for up to 19 months to develop tumors. (**B**) Beta-actin-driven *PAX3-FOXO1* primarily produced primitive neuroectodermal tumors in a wild-type genetic background. Shown for all tumors are representative examples with the presentation of gross morphology and GFP expression patterns coupled with a hematoxylin and eosin stain. (**C**) CMV-driven *PAX3-FOXO1* produced rhabdomyosarcoma in the $tp53^{M214K/M214K}$-sensitized genetic background. (**D**) Ubiquitin-driven *PAX3-FOXO1* produced an undifferentiated sarcoma in a wild-type genetic background. (**E**) RNAseq data from zebrafish *PAX3-FOXO1* fluorescent tumors showing the number of reads supporting the presence of the human fusion-oncogene. (**F**) Tumor incidence of GFP + tumors detected in BetaActin-GFP2A-*PAX3FOXO1* (n = 74) injected zebrafish versus BetaActin-GFP (n = 147) injected controls in a wildtype genetic background. (**G**) Tumor incidence of GFP + tumors detected in CMV-GFP2A-*PAX3FOXO1* (n = 31) injected zebrafish in a $tp53^{M214K}$-sensitizing genetic background.

DOI: https://doi.org/10.7554/eLife.33800.003

The following figure supplements are available for figure 1:

**Figure supplement 1.** Promoter restricted expression of human *PAX3-FOXO1* has different effects on survival in developing zebrafish.

DOI: https://doi.org/10.7554/eLife.33800.004

**Figure supplement 2.** Junctional sequence used to map human *PAX3-FOXO1* fusion RNAseq reads from zebrafish tumors.

*Figure 1 continued on next page*

*Figure 1 continued*

DOI: https://doi.org/10.7554/eLife.33800.005

fusion-oncogene is transforming in zebrafish when expressed in a variety of cellular contexts and that some, but not all, cellular identities are susceptible to transformation.

The beta actin, CMV, and ubiquitin promoter restricting *PAX3-FOXO1* expression have unique latency, penetrance, and spectrum of tumor development (*Figure 1F–G*; *Supplementary file 1*). Tumor incidence curves are shown for the beta actin promoter driving *PAX3-FOXO1* in a wild-type background, and for the CMV promoter driving *PAX3-FOXO1* in the $tp53^{M214K/M214K}$ mutant background (*Figure 1F–G*). Rhabdomyosarcoma only develops in the $tp53^{M214K}$ background, predominantly with the CMV promoter as compared to the beta actin promoter. Since human *PAX3-FOXO1* is active in zebrafish and produces discrete functional readouts, we next applied our model to identify *PAX3-FOXO1* targets that are engaged during development to induce tumorigenesis.

## *PAX3-FOXO1* has a different impact on embryonic development than *PAX3*

The *PAX3-FOXO1* fusion-oncogene functions as an aberrant transcription factor due to the *PAX3* DNA-binding domain being linked to the *FOXO1* transactivation domain (*Linardic, 2008*; *Cao et al., 2010*). To identify activity unique to *PAX3-FOXO1* versus the normal *PAX3* transcription factor, we performed a detailed analysis of the functional effects of their expression during zebrafish development. We utilized a mosaic model system, in which plasmid DNAs containing the beta actin promoter driving GFP, or GFP-tagged *PAX3* or *PAX3-FOXO1* were injected into developing zebrafish using the Tol2 transposon system (*Figure 2A*). The expression of each construct was tracked using a viral 2A linked fluorescent protein, and mosaic integration of the construct can be observed with fluorescence by 24 hr post-injection (*Figure 2B*). During the first three days of life, *PAX3-FOXO1* injected zebrafish exhibited reduced survival as compared to *PAX3* injected zebrafish (*Figure 2C*). Furthermore, *PAX3-FOXO1*-induced unique embryonic phenotypes, including cyclopia, which was present in 30% of injected embryos. To address if cyclopia was due to *PAX3-FOXO1* expression, we co-injected GFP2A-*PAX3FOXO1* with a morpholino that targets the expression construct. In this scenario, the morpholino binds the start codon of GFP and inhibits the ribosome's association with the mRNA transcript, thus inhibiting the translation of the GFP-viral2A-*PAX3FOXO1* mRNA. Morpholino activity was tracked by quantifying the reduced number of GFP positive embryos after injection with titrating doses of morpholino (*Figure 2D*). Co-injection of *PAX3-FOXO1* and a morpholino that knocks-down GFP-*PAX3-FOXO1* expression eliminated the cyclopia phenotype (*Figure 2D*). This is significantly different from the uninjected, GFP injected, and the *PAX3* injected groups, suggesting that *PAX3-FOXO1* is regulating distinct developmental phenotypes.

This discrepancy between *PAX3-FOXO1* and *PAX3* tolerance was evident at the level of individual cells. To demonstrate this, embryos were injected with beta-actin-driven constructs containing: GFP only, GFP-tagged *PAX3,* or GFP-tagged *PAX3-FOXO1*. Injected zebrafish embryos were allowed to develop for 24 hr, and then dissociated to single-cell suspensions. Fluorescent activated cell sorting (FACS) showed that *PAX3-FOXO1* expression significantly reduced the number of GFP + cells at 24 hr post-fertilization as compared to *PAX3* (*Figure 2E*). This reduced viability of *PAX3-FOXO1* + zebrafish was consistent in injected embryos that were raised to adulthood and screened at 3 months of age for detectable fluorescence. In BetaActin-mCherry or GFP fluorescent controls, fluorescence was observed in over 70% of adult zebrafish. In *PAX3*-injected zebrafish, mCherry fluorescence was observed in 32% of adults. We found that only 3% of adult zebrafish had detectable GFP-*PAX3FOXO1* expression with resulting tumor development or asymmetric skeletal muscle (*Figure 2F–G*). This was not true for BetaActin-GFP or BetaActin-GFP-*PAX3*, which do not generate tumors or affect normal development or survival.

One explanation for this disappearance of *PAX3-FOXO1* + cells during the course of development is that *PAX3-FOXO1*-injected zebrafish have a significant increase in the number of cells undergoing apoptosis. At 24 hr post-fertilization, TUNEL staining was performed on embryos injected with either GFP-*PAX3* or GFP-*PAX3FOXO1*. These same embryos were counter-stained for GFP to detect the expression levels of their respective transgenes (*Figure 2H*). *PAX3-FOXO1*-injected

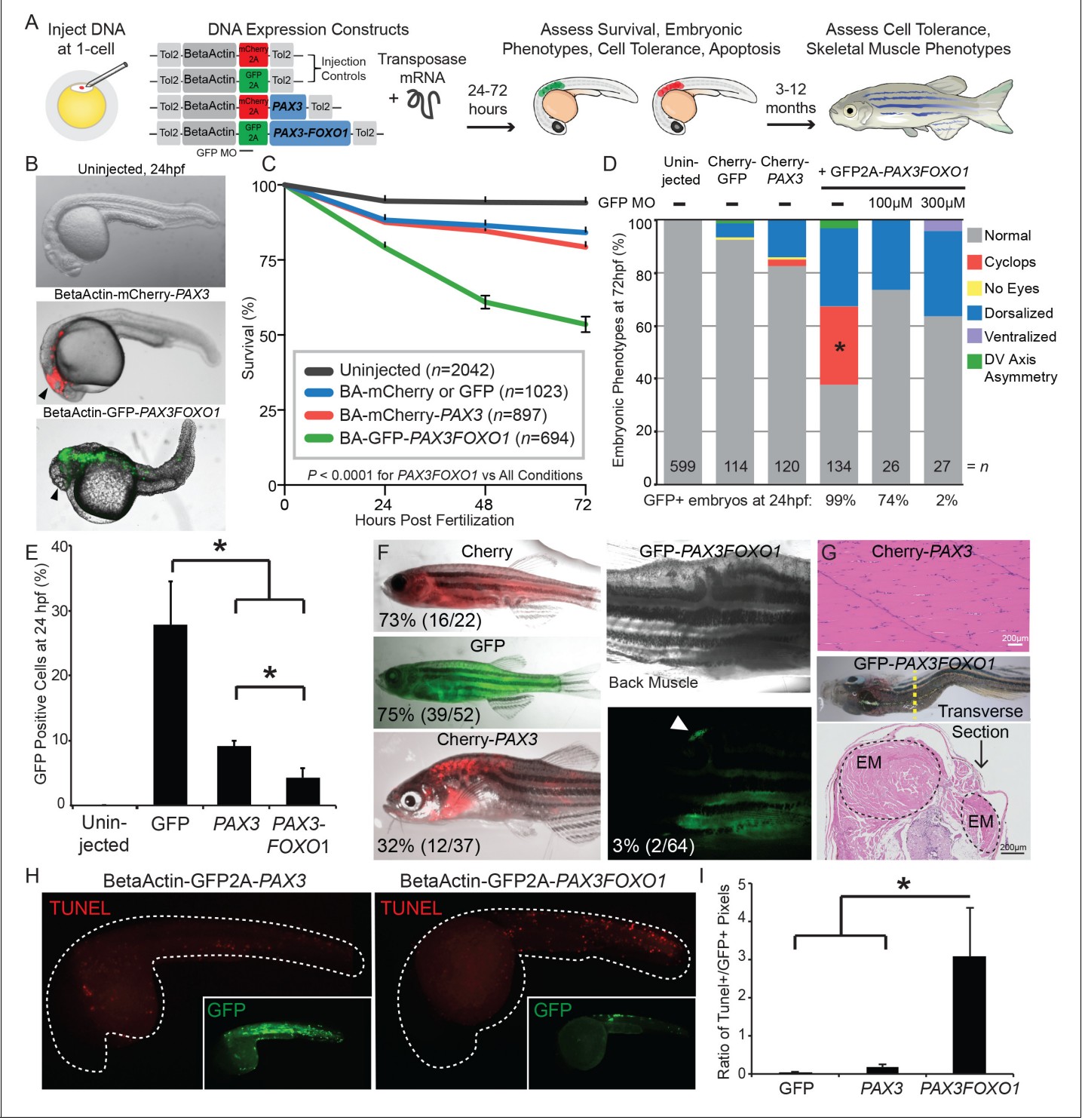

Figure 2. *PAX3* and *PAX3-FOXO1* have distinct impacts and tolerance during embryonic development and in adult zebrafish. (A) Strategy for assessing functional differences of beta-actin-driven *PAX3* and *PAX3-FOXO1* in a vertebrate system. (B) Representative images at 24 hr post-fertilization of Uninjected, mCherry2A-*PAX3*, and GFP2A-*PAX3FOXO1*-injected zebrafish. (C) Survival curve of Uninjected, GFP2A or mCherry2A injected controls, mCherry2A-*PAX3*, and GFP2A-*PAX3FOXO1*. Error bars represent SE. Log rank test, p<0.0001 for *PAX3FOXO1* versus all other conditions. (D) Embryonic phenotypes scored at 3 days post-injection. * indicates p<0.05, for *PAX3 vs PAX3FOXO1*, Fisher's exact test. MO- morpholino. DV- Dorso-Ventral. (E) Percentage of GFP + cells from dissociated zebrafish embryos as quantified by fluorescent activated cell sorting (FACS). Error bars represent SD across three independent experiments. * indicates p<0.05, two-tailed Student's t-test. (F) Adult zebrafish over 3 months of age robustly expressed beta-actin-driven Cherry, GFP, or Cherry2A-*PAX3* and developed normally. Zebrafish injected with BetaActin-GFP2A-*PAX3FOXO1* displayed

*Figure 2 continued on next page*

Figure 2 continued

developmental defects or developed tumors. Arrow denotes GFP + area. The percentage indicates zebrafish with detectable fluorescence at adulthood. (G) Hematoxylin and eosin staining showed normal histology of BetaActin-*PAX3* expressing skeletal muscle (sagittal section) at 299 days of age, and abnormal histology of BetaActin-*PAX3FOXO1* epaxial muscle exhibiting dramatic left-right asymmetry (transverse section, asymmetry of left-right epaxial muscle (EM) marked by dotted lines) at 307 days of age. Scale bars, 200 microns. EM- epaxial muscle. (H) Representative images from zebrafish embryos injected with GFP2A-*PAX3* and GFP2A-*PAX3FOXO1* that are fixed at 24 hr post-injection and then TUNEL performed (rhodamine). Embryos were counter-stained for GFP to indicate transgene expression. (I) Quantification of TUNEL-positive pixels normalized to GFP positive pixels, indicated a higher proportion of *PAX3-FOXO1* cells were undergoing apoptosis. Error bars represent SD, *n* = 6–8 embryos per group, * indicates p<0.05, two-tailed Student's t-test.

DOI: https://doi.org/10.7554/eLife.33800.006

The following figure supplements are available for figure 2:

**Figure supplement 1.** The *tp53*$^{M214K}$ mutation modifies the CMV-*PAX3FOXO1* phenotype in developing zebrafish.

DOI: https://doi.org/10.7554/eLife.33800.007

**Figure supplement 2.** A model for *tp53*$^{M214K}$ mediation of CMV-*PAX3FOXO1* RMS tumorigenesis.

DOI: https://doi.org/10.7554/eLife.33800.008

embryos had an increase in the number of TUNEL-positive cells undergoing apoptosis as compared to GFP controls and *PAX3* in wild-type zebrafish (*Figure 2I*). Given these data and the propensity to generate RMS we mechanistically evaluated the contribution of *tp53* mutations to early *PAX3-FOXO1*-induced apoptosis (*Figure 1C and G*; *Figure 2—figure supplement 1A*). We found that *tp53*$^{M214K}$ mutant zebrafish embryos are unable to invoke an appropriate response to *PAX3-FOXO1* + cells, resulting in a gross reduction in survival by 3 days of age (*Figure 2—figure supplement 1B*). Further, injected CMV-*PAX3FOXO1* + cells are more abundant in *tp53*$^{M214K}$ injected mutant zebrafish as compared to their wild-type counterparts by 28 hr post fertilization (*Figure 2—figure supplement 1C–D*). This significant increase in the number of GFP-*PAX3FOXO1* + cells is coupled with a reduction in the ratio of TUNEL-positive cells, indicating that *tp53*$^{M214K}$ mutant zebrafish provide a susceptible environment for *PAX3-FOXO1* + cellular persistence and ultimately tumor development (*Figure 2—figure supplement 1E–G*; *Figure 2—figure supplement 2*). This dichotomy of cellular tolerance and apoptosis is not true for wildtype or *tp53*$^{M214K}$ mutant zebrafish injected with GFP controls, or GFP-*PAX3* (*Figure 2—figure supplement 1B–G*). Therefore, at both an organismal and individual cell level, ectopic expression of *PAX3-FOXO1* was less tolerated than the normal *PAX3* gene. Understanding the exact mechanisms for cell tolerance, and defining these discrepancies, could identify the earliest events in RMS transformation or alternatively tumor suppressive responses.

## *her3* is a novel developmental target of human *PAX3-FOXO1*.

Given that *PAX3-FOXO1* is uniquely tumorigenic and has different embryonic effects than *PAX3* this warranted a more thorough study of the specific developmental pathways and targets that were mediating these early outcomes. To accomplish this, we injected zebrafish embryos with DNA expression constructs in which the beta actin promoter drives either a GFP control, or GFP-tagged *PAX3* and *PAX3-FOXO1*, and allowed the embryos to develop for 24 hr. The GFP + cell population from zebrafish embryos was then FACS sorted, total RNA isolated, and microarrays were used to determine differential gene expression signatures. After comparing differentially expressed genes versus the GFP injected control for *PAX3* and *PAX3-FOXO1*, the most highly enriched Gene Ontology (GO) terms indicated that *PAX3* and *PAX3-FOXO1* act as transcription factors (*Figure 3A*). This was expected given their well-documented roles in the literature and suggested that the mammalian forms are active in zebrafish systems. Furthermore, DAVID analysis of the enriched genes during normal development suggested that these sorted cell populations identify different subsets of zebrafish embryonic tissues. *PAX3*-positive cells were most indicative of the neural plate, neuroectoderm, or ectoderm, whereas *PAX3-FOXO1* was indicative of a somite, segmental plate, or optic vesicle and immature eye (*Figure 3B*). Even though *PAX3* and *PAX3-FOXO1* were injected under the control of the same promoter, cells expressing these genes represent either (1) different cell populations or (2) identical cell populations that have differential transcriptional responses as early as 24 hr post-fertilization.

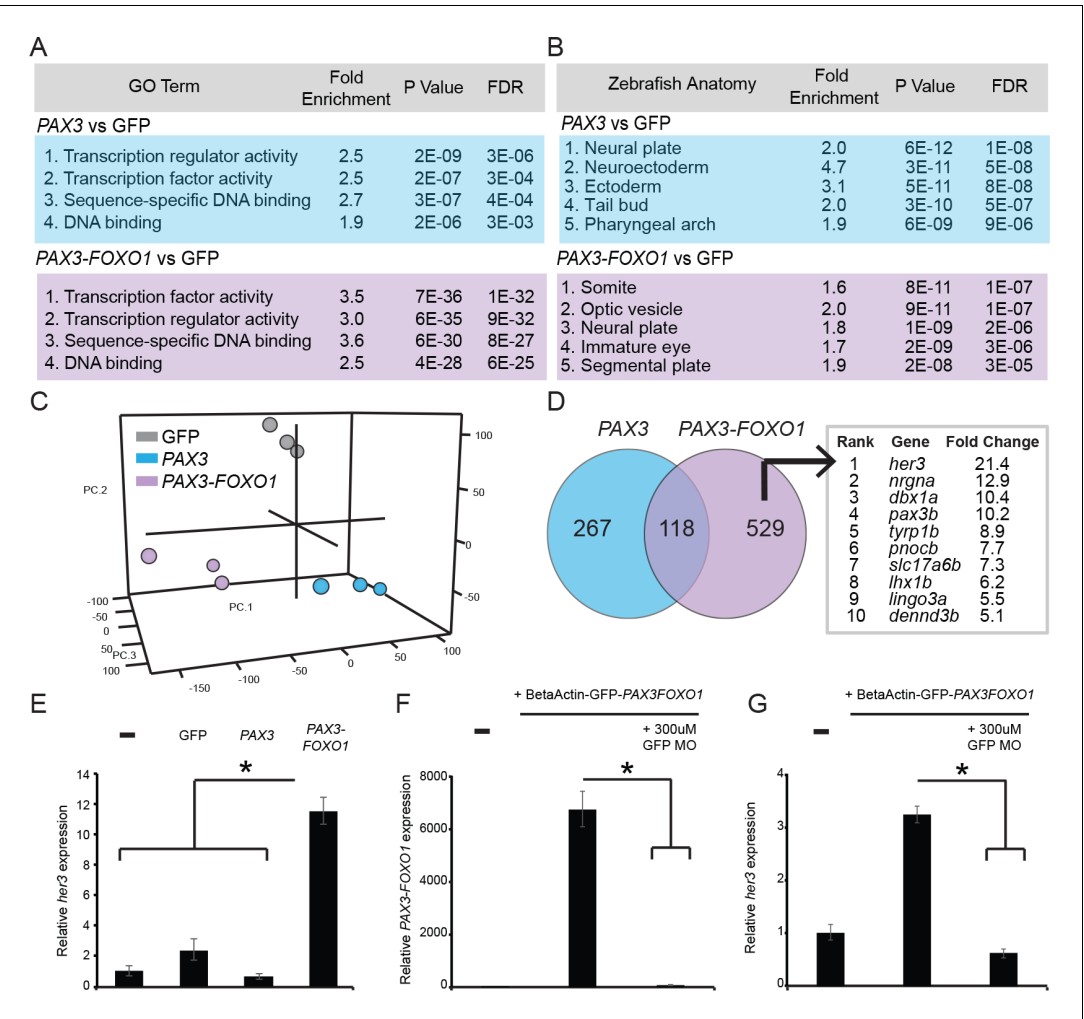

**Figure 3.** *PAX3* and *PAX3-FOXO1* induce distinct gene expression signatures during development, identifying a unique *PAX3-FOXO1* target, *her3*. Microarray analysis identified differentially expressed gene signatures in 24 hr old zebrafish FACS sorted embryonic cells that expressed either GFP2A-*PAX3FOXO1* or GFP2A-*PAX3*. (**A**) Gene ontology terms associated with *PAX3* or *PAX3-FOXO1*. (**B**) Embryonic tissues indicative of *PAX3* or *PAX3-FOXO1* gene sets. (**C**) 3D principal components analysis (PCA) of GFP injected controls, *PAX3*, and *PAX3-FOXO1*. (**D**) Intersection of up-regulated genes for *PAX3* and *PAX3-FOXO1* as compared to GFP controls. Uniquely up-regulated *PAX3-FOXO1* target genes were included in downstream analysis only if they possessed a human ortholog. Shown are genes rank ordered based on the fold-change of their expression. (**E**) qRT-PCR of *her3* levels from 24-hr-old zebrafish embryos that are either uninjected controls or injected with GFP, *PAX3*, or *PAX3-FOXO1*. (**F**) qRT-PCR for *PAX3-FOXO1* mRNA levels from 24-hr-old zebrafish embryos injected with *PAX3-FOXO1*, or *PAX3-FOXO1* in combination with a GFP morpholino (GFP MO) that inhibits transgene expression. (**G**) Same samples as in F, but qRT-PCR was performed for *her3*. In E-G the SD is derived from technical triplicates, * indicates p<0.05, two-tailed Student's t-test.

DOI: https://doi.org/10.7554/eLife.33800.009

We then performed a principal component analysis (PCA) and found that *PAX3* and *PAX3-FOXO1* expression changes are distinct and show statistical significance in a developmental context (*Figure 3C*). This does not contradict the fact that there are a large number of expression changes that are similar between these two transcription factors. Therefore, unique developmental and oncogenic targets of *PAX3-FOXO1* could be ascertained by directly comparing its activity to that of *PAX3*. To determine what these *PAX3-FOXO1* developmental targets might be, we looked at the overlap of induced genes for *PAX3* or *PAX3-FOXO1* versus the GFP controls. We found that *PAX3* and *PAX3-FOXO1* up-regulated 118 shared genes, whereas *PAX3-FOXO1* uniquely induced 529

genes. These 529 genes were of interest because of their association with the fusion-oncogene and potentially tumorigenesis. Of these 529 genes, those without a human ortholog were eliminated from further analysis due to our interest in human disease. This produced a list of primarily developmental transcription factors and homeobox genes, including *her3*, *pax3b*, *dbx1a*, and *lhx1b*. The most highly induced gene unique to *PAX3-FOXO1* expression was *her3*, with a 21-fold increase as compared to the GFP control (*Figure 3D*).

We validated that *her3* expression was induced by the human *PAX3-FOXO1* fusion-oncogene in zebrafish by injection of GFP control, *PAX3*, and *PAX3-FOXO1*. Embryos were allowed to develop and then total RNA was isolated from an embryo pool, cDNA transcribed, and a qRT-PCR performed for *her3* expression levels. This analysis indicated that *her3* induction was unique to *PAX3-FOXO1* (*Figure 3E*). To verify that the up-regulation of *her3* was dependent on *PAX3-FOXO1*, we co-injected the BetaActin-*PAX3FOXO1* construct with a morpholino that inhibits its early expression by targeting the GFP2A-*PAX3FOXO1* transcript. At 24 hr of age, *PAX3-FOXO1* mRNA levels were measured by qRT-PCR, and co-injection of the morpholino significantly knocked down *PAX3-FOXO1* expression (*Figure 3F*). A qRT-PCR was performed on these same samples to evaluate *her3* expression levels, and the results replicated those of *PAX3-FOXO1* (*Figure 3G*). These data suggest that *her3* is a novel target of *PAX3-FOXO1 in vivo*, that *her3* is not induced by *PAX3*, and that *her3* expression is dependent on the *PAX3-FOXO1* oncogene.

### *HES3*, the human ortholog of *her3*, inhibits in vivo myogenesis and supports inappropriate persistence of *PAX3-FOXO1* + cells

A hallmark of rhabdomyosarcoma is that tumors express early markers along the skeletal muscle lineage such as MYOD1, MYOG and Desmin; however, these tumors fail to terminally differentiate indicating a developmental arrest (*Parham and Ellison, 2006*; *Saab et al., 2011*). To determine if this was recapitulated in our zebrafish model, we performed a mosaic co-injection strategy with the beta actin promoter driving GFP-mCherry controls, human *HES3* alone, human *PAX3-FOXO1* alone, or a co-injection of *PAX3-FOXO1* and *HES3* (*Figure 4A*). Co-injection of transgenes linked to highly expressed promoters generated mosaic integration that overlaid with muscle proteins such as myosin by 24 hr post-fertilization (*Figure 4B*). Based on this observation, zebrafish were injected and allowed to develop for 24 hr, and then fluorescent embryos harvested and total RNA isolated and reverse transcribed for qRT-PCR analysis of muscle marker genes. This survey of temporally stereotyped muscle markers included early markers *myod* and *myog* and terminal muscle differentiation markers *myl1* and *myhz2*. We found that *PAX3-FOXO1* did not affect the mRNA levels of *myod* or *myog*; however, by 24hpf there was a significant reduction in the expression of *myl1* and *myhz2*. Surprisingly, overexpression of *HES3* alone had a similar effect of inhibiting terminal muscle differentiation, indicating that *her3*/*HES3* contribute to maintaining a more primitive cellular state. Together, *PAX3-FOXO1* and *HES3* had no significant additive effect, suggesting they are functioning in a linear pathway with *her3*/*HES3* being downstream of *PAX3-FOXO1* (*Figure 4C*). Hence, *PAX3-FOXO1* and *HES3* inhibit differentiation in vivo in a vertebrate system via a shared mechanism.

The majority of the *PAX3-FOXO1* and *HES3* co-injected cells overlaid indicating that this mosaic system is a powerful model to study the interactions of two genes within the same cell (*Figure 4A and D*). Given this capacity, we next determined if *HES3* modified the behavior of GFP-*PAX3FOXO1* + cells during embryogenesis. Zebrafish embryos were co-injected at the single-cell stage with two independent plasmids that were dually integrated into the genome using the Tol2-transposon-based system. The conditions included the beta actin promoter driving: (1) mCherry and GFP, (2) GFP and mCherry-*HES3*, (3) mCherry and GFP-*PAX3FOXO1*, and (4) GFP-*PAX3FOXO1* and mCherry-*HES3*. Both GFP and mCherry images were taken using the same settings at both 24 hr post-fertilization and 72 hr post-fertilization to evaluate the capacity of the cells to survive (*Figure 4D*). The number of GFP and mCherry positive pixels were calculated independently and plotted for both timepoints (*Figure 4E–F*). Co-injection of *HES3* allowed *PAX3-FOXO1* cells to survive and/or proliferate, with a greater number of GFP-positive pixels and thus *PAX3-FOXO1* positive cells persisting by 72hpf (*Figure 4E*). This phenomenon was unique to the co-injection of *HES3* + *PAX3FOXO1*, as the number of pixels that were positive between 24hpf and 72hpf for mCherry + GFP-*PAX3FOXO1* was not significantly different. Additionally, the number of mCherry positive pixels from 24hpf to 72hpf was insignificantly different for mCherry-*HES3* + GFP-*PAX3-FOXO1* indicating that this observation was not globally applicable, but unique to the persistence of

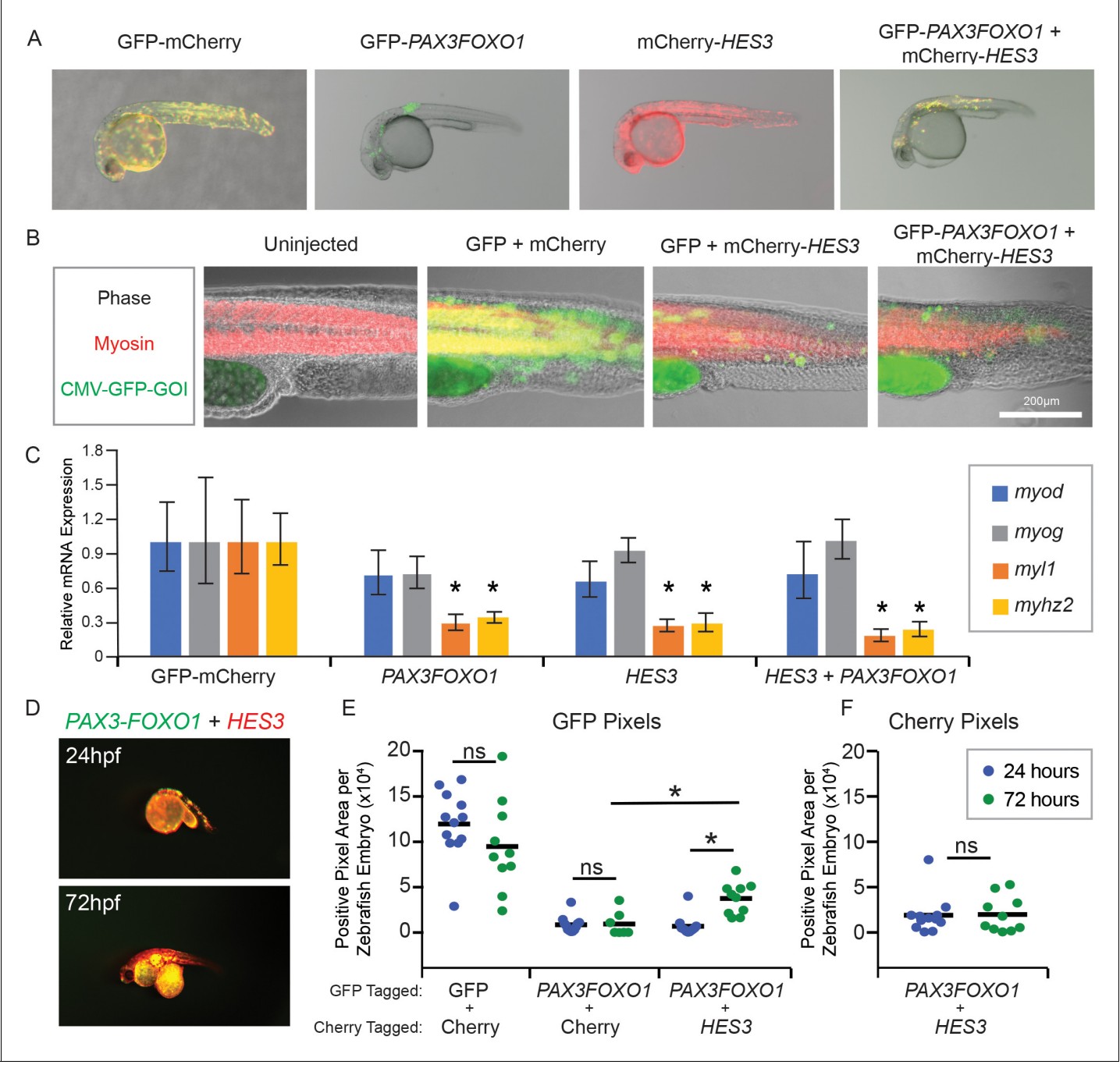

**Figure 4.** *HES3* inhibits myogenic differentiation in developing zebrafish and supports persistence of *PAX3-FOXO1*-positive cells. (**A**) Zebrafish embryos were injected at the single-cell stage with the beta actin promoter driving GFP-Cherry, GFP-*PAX3FOXO1*, mCherry-*HES3*, or combined mCherry-*HES3* and GFP-*PAX3FOXO1*. Shown are representative embryos at 24 hr post fertilization with indicated transgene expression. (**B**) Representative overlays of zebrafish embryo musculature that were fixed at 24 hr and immunofluorescence performed for myosin (red) and injected genes of interest (GOI; green). (**C**) Pools of *n* = 5 embryos were harvested at 24 hr and markers of myogenesis assessed by qRT-PCR, including *myod*, *myog*, *myl1*, and *myhz2*. SD is derived from technical triplicates. * indicates significant differences between treatment group and the GFP-mCherry control at a threshold of p<0.05, two-tailed Student's t-test. (**D**) Representative overlay of images from co-injections of mCherry-*HES3* and GFP-*PAX3FOXO1* from the same embryo at 24 and 72 hr post-fertilization. Images were taken with the same exposure settings and objective. (**E**) Quantification of the number of positive pixels for each embryo imaged at 24 and 72 hr post-fertilization. GFP-positive pixels are plotted after the same settings are applied for imaging and analysis. Each marker represents a single zebrafish embryo at 24 or 72 hr post fertilization, *n* = 6–12 embryos per group. Black bar is the mean, and * indicates p<0.05, two-tailed Student's t-test. ns- not significant. (**F**) Same analysis as in E but for mCherry positive pixels.

DOI: https://doi.org/10.7554/eLife.33800.010

*Figure 4 continued on next page*

*Figure 4 continued*

The following figure supplements are available for figure 4:

**Figure supplement 1.** *HES3* facilitates cellular tolerance of CMV-*PAX3FOXO1* expression in developing zebrafish but does not alleviate the apoptosis phenotype.

DOI: https://doi.org/10.7554/eLife.33800.011

**Figure supplement 2.** A model for *HES3* facilitation of CMV-*PAX3FOXO1* embryonic cellular persistence.

DOI: https://doi.org/10.7554/eLife.33800.012

*PAX3-FOXO1* (*Figure 4F*). These results suggest that *HES3* promotes a more tolerant cellular environment allowing for survival with expression of the highly toxic *PAX3-FOXO1*.

One hypothesis is that the co-expression of *HES3* allows for *PAX3-FOXO1* expressing cells to circumvent apoptosis. By implementing our CMV injected mosaic zebrafish models, we found that *HES3* overexpression in zebrafish allows for *PAX3-FOXO1*+ cells to either inappropriately persist or divide, resulting in an increased number of observable GFP-*PAX3FOXO1* + pixels by 24 hr post-fertilization (*Figure 4—figure supplement 1A–C*). We performed TUNEL assays on these same zebrafish embryos and found that *HES3* alone does not induce apoptosis. Further, there is no significant difference in the normalized ratio of TUNEL cells for *PAX3-FOXO1*-injected embryos as compared to *PAX3-FOXO1* + *HES3* (*Figure 4—figure supplement 1D–E*). These data indicate that *PAX3-FOXO1* + *HES3* cooperation facilitates *PAX3-FOXO1* + cells to persist during embryogenesis independently of apoptosis inhibition at this timepoint (*Figure 4—figure supplement 2*).

## *HES3* inhibits myogenesis in mammalian myoblasts

Impaired skeletal muscle differentiation is a signature feature of human RMS tumors. To evaluate how *HES3* is contributing to oncogenesis in fusion-positive RMS, we determined its capacity to inhibit differentiation in mammalian cell culture (*Figure 5A*). Stable mouse C2C12 myoblast cell lines were generated by transfection and selection of CMV-*HES3* or CMV-Empty control, and the resulting cell lines were evaluated for overexpression of *HES3* by qRT-PCR (*Figure 5B*). Myoblasts were viable with *HES3* overexpression and were next tested to determine if there was a temporal or functional difference in their myogenic capacity. Cells were seeded onto porcine gelatin coated plates and collected at one and 6 days post plating after being exposed to low-serum concentrations and supplemented with insulin to promote fusion. The expression levels of selected genes were evaluated to represent the spectrum of myogenic differentiation, including *MyoD*, *MyoG*, *Myl1*, and *Myh1* (*Figure 5A*). By day 6 of differentiation, *MyoD*, *MyoG*, *Myl1*, and *Myh1* exhibited significant inhibition of expression in C2C12-*HES3* overexpressing cells as compared to the C2C12-Empty controls. Moreover, *Myl1* and *Myh1*, which represent terminal muscle differentiation, were the most affected by *HES3* overexpression at fusion day 6, with a 2.9–3.2 fold decrease in expression levels, as compared to a 1.8–1.9 fold decrease observed for *MyoD* and *MyoG* (*Figure 5B*).

A decrease in the mRNA expression of markers of muscle identity translated to a reduction in the functional capacity to fuse into multi-nucleated myofibers. C2C12-*HES3* overexpressing cells and C2C12-Empty controls were differentiated in low-serum conditions, and immunofluorescence performed to determine protein expression and localization of myosin heavy chain (*Figure 5C*). Both C2C12-Empty and C2C12-*HES3* cells exhibited the capacity to fuse into multi-nucleated myotubes that express myosin heavy chain upon terminal differentiation. However, there was a stark difference in the success of the differentiation. After analyzing the percentage of myogenic nuclei, or the nuclei within a myofiber, there was 2.3X more myogenic nuclei in C2C12-Empty as compared to the C2C12-*HES3* overexpressing cells (*Figure 5D*). Furthermore, the fusion index (myogenic nuclei with > 3 nuclei present per myofiber) indicated a significant reduction in the fusion capacity of *HES3* overexpressing cells (*Figure 5E*). HES3 overexpression also modified the kinetics of C2C12 differentiation, with myosin protein being undetectable at day 4 of fusion in C2C12-HES3 overexpressing cells and detectable in C2C12 controls (*Figure 5—figure supplement 1A–C*). These contrasting fusion capacities cannot be fully attributed to a dominant negative effect on MyoD and inhibition of MyoD expression during fusion initiation. In fact, there were comparable levels of MyoD protein at day 3 of C2C12 fusion. By day 9 there is a trend towards decreased MyoD in HES3 overexpressing

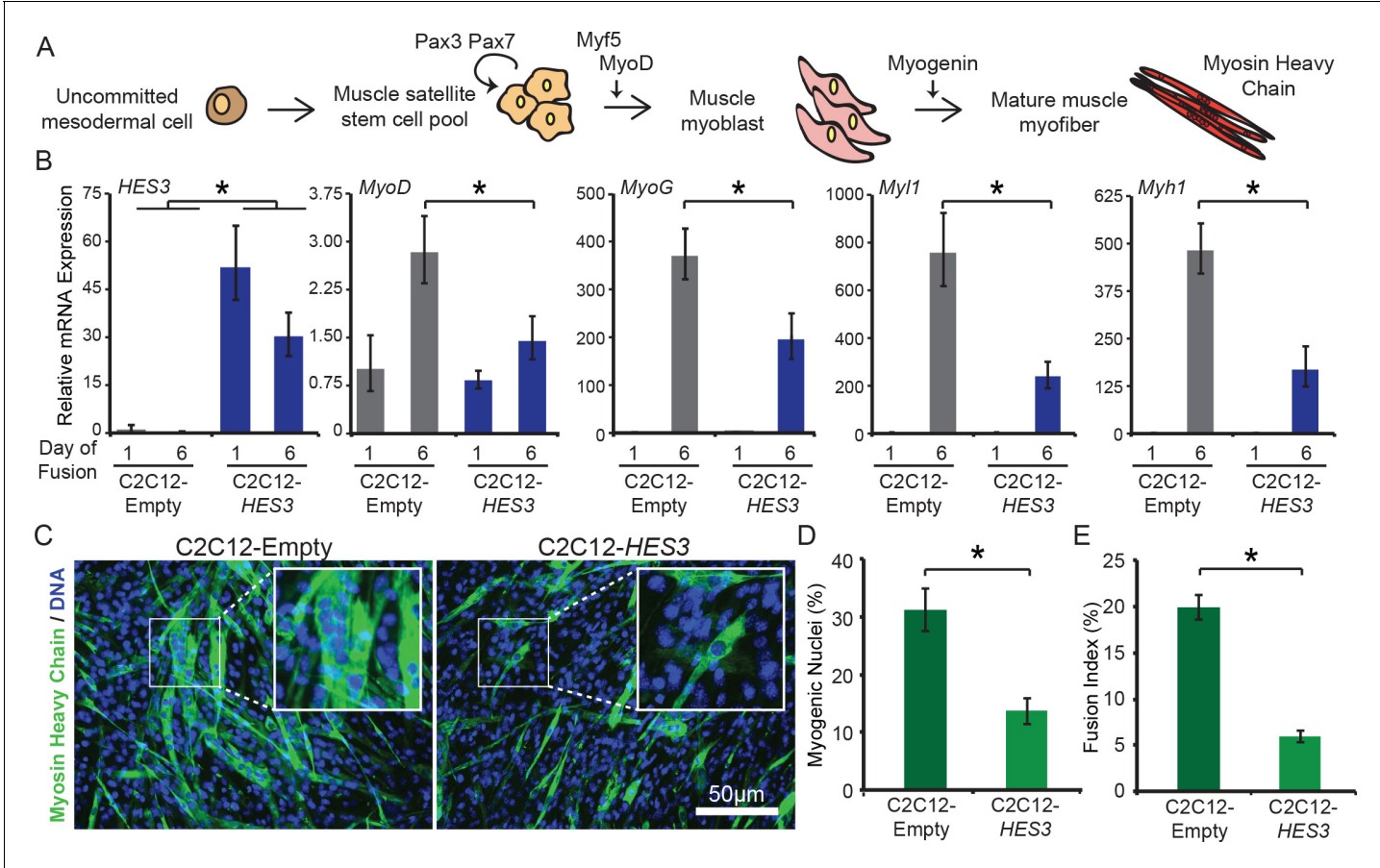

**Figure 5.** *HES3* overexpression inhibits myogenic differentiation of mouse muscle myoblasts. (**A**) Schematic of muscle development. Specifically noted are key skeletal muscle transcription factors and proteins assessed in B. Muscle satellite cells that self-renew express *Pax3*/Pax7. During development, they are activated to further differentiate by expression of MyoD, the master muscle regulator, after which they become myoblasts, an intermediate cell type. Myoblasts proliferate and fuse together to form multi-nucleated and contractile myofibers that express structural proteins, such as myosin heavy chain. (**B**) C2C12-Empty and C2C12-*HES3* overexpressing cells were seeded in growth media and after 24 hr exposed to differentiation media. At this point Day 1 of fusion was collected. Following 5 additional days in fusion media the final timepoint was collected. qRT-PCR was performed for *HES3*, and the following muscle marker genes: *MyoD*, *MyoG*, *Myl1*, *Myh1*. SD is derived from technical triplicates. * indicates $p < 0.05$, two-tailed Student's t-test. (**C**) C2C12-Empty and C2C12-*HES3* cells were differentiated for five days, fixed, and immunofluorescence performed for Myosin Heavy Chain (MyHC) protein. Cells were counterstained with DAPI to detect DNA. Shown are representative images of the fusion capacity for each condition. (**D**) Quantification of the differentiation capacity of *HES3* overexpressing mouse myoblasts. Plotted are the number of myogenic nuclei (# of nuclei within a MyHC + cell divided by the total nuclei) with $n = 3$ technical replicates per group ± SD. Experiment was repeated with biological replicates. * indicates $p < 0.05$, two-tailed Student's t-test. (**E**) Same data as in D represented as the fusion capacity which focuses on multi-nucleation. The fusion capacity was calculated by including nuclei with $n > 3$ nuclei per MyHC + cell divided by the total number of nuclei. Plotted is the mean of $n = 3$ technical replicates per group ± SD. Experiment was repeated with biological replicates. * indicates $p < 0.05$, two-tailed Student's t-test.

DOI: https://doi.org/10.7554/eLife.33800.013

The following figure supplements are available for figure 5:

**Figure supplement 1.** HES3 overexpression inhibits the myogenic differentiation kinetic in mouse muscle myoblasts.
DOI: https://doi.org/10.7554/eLife.33800.014

**Figure supplement 2.** HES3 does not alter MyoD expression during fusion initiation but inhibits MyoD during terminal differentiation.
DOI: https://doi.org/10.7554/eLife.33800.015

C2C12 myotubes (*Figure 5—figure supplement 2A–C*). Hence, HES3 is inhibiting the differentiation process and promoting a more primitive cellular state in mammalian systems.

## HES3 overexpression increases pro-tumorigenic features in cell culture systems

To determine if *HES3* overexpression confers properties indicative of tumorigenic capacity in cell culture systems, we analyzed *HES3*'s impact on C2C12 mouse myoblasts and human rhabdomyosarcoma cells. We focused on the human cell line, Rh30, which is derived from a bone marrow metastasis of ARMS and contains the *PAX3-FOXO1* fusion (*Hinson et al., 2013*). Stable Rh30 cell lines were generated by transfection and selection of either the CMV-Empty or CMV-*HES3* expression construct. A qRT-PCR verified *HES3* overexpression and that *PAX3-FOXO1* was present (*Figure 6A–B*). This overexpression strategy was pursued because of the low levels of baseline *HES3* in Rh30. We examined a panel of rhabdomyosarcoma cell lines and have observed low *HES3* levels by RNAseq, and no detectable endogenous HES3 protein expression with commercially available antibodies (data not shown). To determine if *HES3* modified the proliferation kinetics of Rh30, cells were plated at a low density and timepoints evaluated from days 1 to 6. We found that in this context *HES3* accelerated the cellular accumulation of Rh30 (*Figure 6C*). Additionally, overexpression of *HES3* modified the expression of a panel of genes implicated in metastasis and more aggressive disease. Notably, there was > 300 fold down-regulation of Missing in Metastasis (*MTSS1*), and an up-regulation of matrix metallopeptidases 3 and 9 (*MMP3* and *MMP9*) (*Figure 6D*).

Complementary results were obtained when C2C12 cells were transfected with either an empty control vector or CMV-*HES3*. Stable C2C12-*HES3* cells overexpressed *HES3* as compared to the C2C12-Empty control (*Figure 6E*). *HES3* was then evaluated for its ability to modulate cellular accumulation. *HES3* significantly accelerated the growth kinetics of C2C12 cells when plated at sub-confluent levels over the course of 6 days (*Figure 6F*). These same cells were plated in suspension in soft agar and cultured for one month to determine the colony formation capacity. *HES3* overexpression significantly increased the number of colonies that formed (*Figure 6G*). This is complemented by data indicating that *HES3* alters the expression of genes that are *PAX3-FOXO1* direct targets or whose expression is modified by *PAX3-FOXO1*; including, *MyoG*, *Igf2*, and *Hoxc6* (*Figure 6H*) (*Khan et al., 1999*; *Lagha et al., 2010*). Potentially, *HES3* is acting as a transcriptional activator and repressor, consistent with its role in normal development.

## HES3 is overexpressed in patient tumors, predicts reduced survival, and identifies potential therapeutic targets

We evaluated the clinical significance of *HES3* by determining *HES3* expression levels in human RMS tumors. We analyzed RNA-Seq and HuEx array data from three independent RMS tumor cohorts, together representing the largest collection of over 200 sequenced RMS tumors (*Chen et al., 2013*; *Shern et al., 2014*). We found that *HES3* had higher expression levels in fusion-positive RMS as compared to fusion-negative RMS (*Figure 7A–C*). Significantly, in RMS patients, the overexpression of *HES3* is associated with reduced overall survival in *PAX3/7-FOXO1* fusion-positive RMS patients, but not in fusion-negative RMS patients (*Figure 7D*). RMS tumors from *Shern et al. (2014)* were evaluated for *HES3* expression levels, and then differentially expressed genes expounded between *HES3* high and low tumors to identify molecular targets with potential translational benefit. A subset of clinically relevant kinases and molecular targets was identified as being concordantly overexpressed with *HES3*; namely, *FGFR4, ALK,* and *PARP1* (*Figure 7E*). Taken together, these data are consistent with our results showing that *her3/HES3* is a *PAX3-FOXO1* target, and that *HES3* contributes to a more aggressive phenotype in rhabdomyosarcoma.

## Discussion

Despite the identification of the primary oncogenic driver *PAX3/7-FOXO1* over 20 years ago, there have been few changes to the primary treatment and prognosis of the disease (*Barr et al., 1993*; *Galili et al., 1993*; *Shapiro et al., 1993*; *Davis et al., 1994*). Developing complementary ARMS animal models will aid in the understanding of disease biology and identify new therapeutic avenues. Zebrafish ARMS models provide a relevant vertebrate developmental context with experimental advantages such as lineage tracing of the ARMS cell of origin, identifying *PAX3-FOXO1* transcriptional targets, and providing a platform for a high-throughput drug screens. Here, we describe a genetic zebrafish model of human disease, in which the human fusion-oncogene, *PAX3-FOXO1*, is

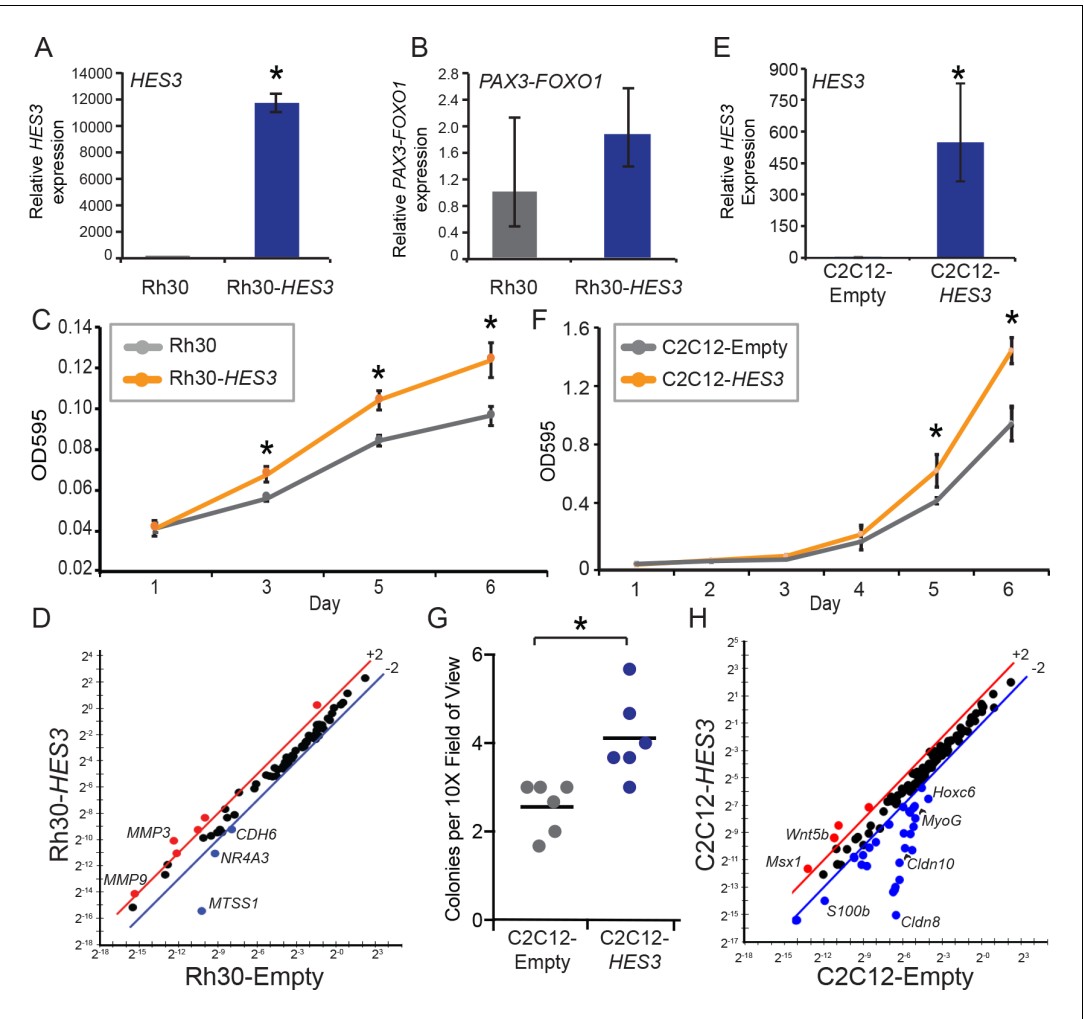

**Figure 6.** *HES3* overexpression promotes pro-tumorigenic features in *PAX3-FOXO1* RMS patient cells and mouse myoblasts. (**A**) Rh30 cells stably express CMV-*HES3* as evaluated by qRT-PCR. SD is derived from technical triplicates. (**B**) *PAX3-FOXO1* levels were assessed by qRT-PCR on the same samples in A. SD is derived from technical triplicates. (**C**) Cells were seeded at a low density and timepoints taken on Days 1, 3, 5, 6 to assess cellular accumulation. At each timepoint, cells were fixed and stained with crystal violet. Plotted is the absorbance from each timepoint with *n* = 4 technical replicates ± SE. Each experiment was repeated two times with biological duplicates. (**D**) qPCR array of metastasis associated genes for Rh30-*HES3* versus Rh30-Empty. Red indicates > 2-fold change above the mean, blue indicates > 2-fold change below the mean. (**E**) C2C12 cells stably expressed CMV-Empty or CMV-*HES3*, and qRT-PCR confirmed *HES3* overexpression. SD is derived from technical triplicates. (**F**) C2C12 cells were seeded at a low density and timepoints taken on Days 1–6 to assess cellular accumulation. At each timepoint, cells were fixed and stained with crystal violet. Plotted is the absorbance from each timepoint with *n* = 4 technical replicates ± SD. Each experiment was repeated two times with biological duplicates. (**G**) Soft-agar colony formation assay was performed over 30 days for C2C12-Empty versus C2C12-*HES3*. Each data point is the mean of three images per well, across six technical replicates. Black bar indicates the mean. (**H**) qPCR array of rhabdomyosarcoma associated genes for C2C12-*HES3* versus C2C12-Empty. Red indicates > 2-fold change above the mean, blue indicates > 2-fold change below the mean. In all panels * indicates $p < 0.05$, two-tailed Student's t-test.

DOI: https://doi.org/10.7554/eLife.33800.016

mosaically integrated into the zebrafish genome. We found that the human fusion is oncogenic in zebrafish under the control of the beta actin, CMV, and ubiquitin promoters, resulting in PNETs, RMS, and histologically undifferentiated sarcoma. Surprisingly, *PAX3-FOXO1* transformation capacity is not limited to myogenic cells, reflecting the lineage specific control of *PAX3* which is relevant in

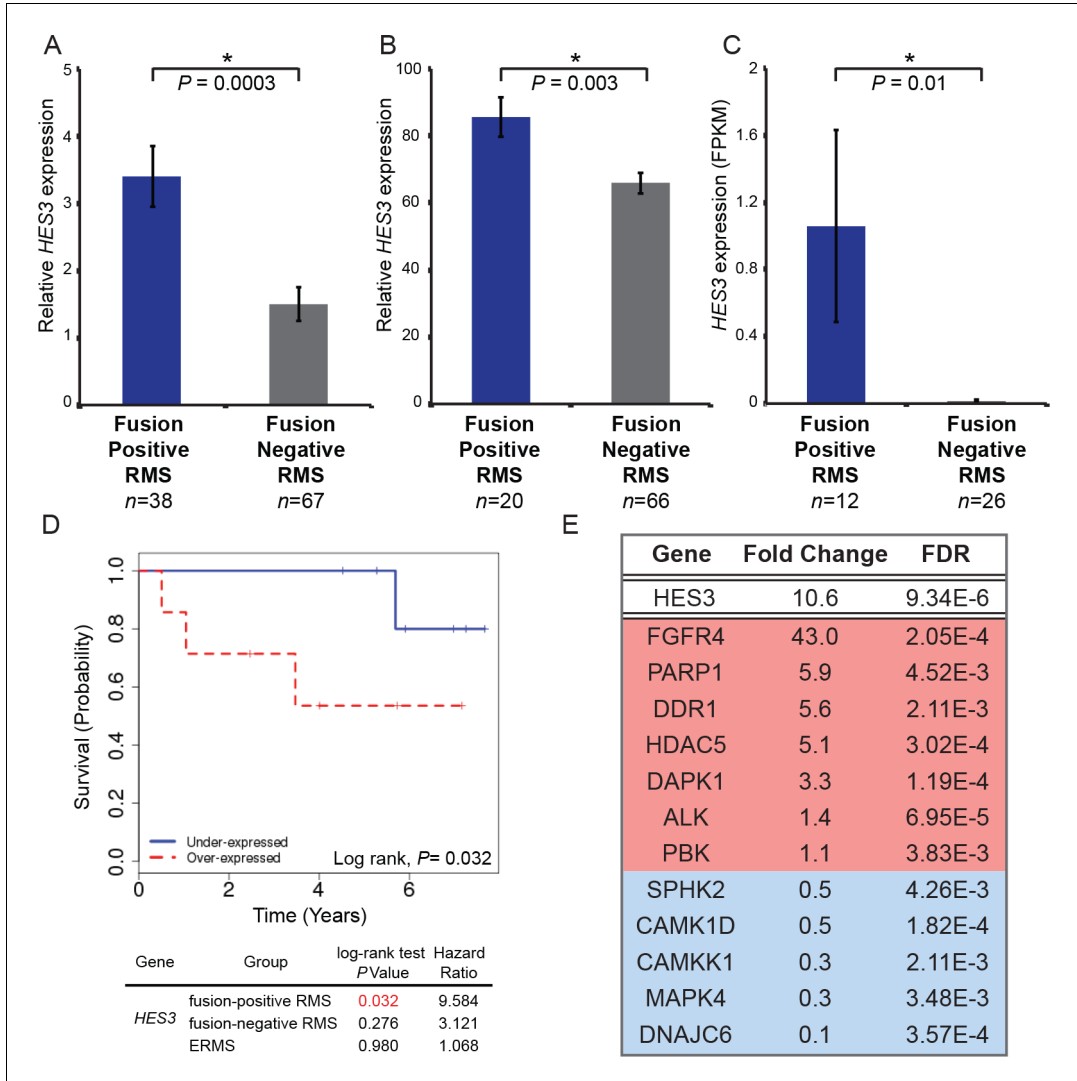

**Figure 7.** *HES3* is overexpressed in fusion-positive RMS patient tumors, predicts reduced overall survival, and identifies potential therapeutic targets. *HES3* expression levels determined in fusion-positive ARMS versus fusion-negative ARMS/ERMS. (**A**) RNAseq data from *Shern et al. (2014)* . (**B**) HuEx data from Triche T, Skapek S; GEO: Accession Number GSE114621. (**C**) RNAseq data from *Chen et al. (2013)*. p-Values were calculated based on two-tailed Student's t-test (**D**) Survival curve of fusion-positive ARMS in the context of *HES3* expression status (*n* = 7 per group). *HES3* expression is derived from the HuEx data presented in B. (**E**) RMS patient tumors from *Shern et al. (2014)* were stratified based on high (FPKM > 1, *n* = 25) or low (FPKM < 1, *n* = 80) *HES3* expression and these subsets evaluated for the most differentially expressed genes (FDR < 0.005). Listed are a subset of kinases or other druggable targets that are up-regulated (red) or down-regulated (blue) in the context of *HES3* overexpression. Genomic datasets from analysis of human subjects are dbGaP controlled-access data, available to users submitting an approved data access request through the dbGaP Authorized Access System (https://dbgap. ncbi.nlm.nih.gov/aa/wga.cgi?page=login.
DOI: https://doi.org/10.7554/eLife.33800.017

the developing brain and muscle. Specific lineages that have been postulated as possible RMS tissues of origin, such as endothelial, neural crest and differentiated muscle (represented here by the *fli1*, *mitfa*, and *unc503* promoters) did not develop tumors, highlighting the requirement for high-expressing ubiquitous promoters in this model. Whereas in patient genomic data, *PAX7-FOXO1* undergoes copy number gain to increase its expression levels, this is not the case for *PAX3-FOXO1*. Likely, *PAX3-FOXO1* must increase its expression level for transformation via transcriptional upregu-lation (*Barr et al., 1996*). To achieve these higher levels of transcription and transformation, we

opted to utilize high-level promoters. This threshold was likely not reached with the *fli1*, *mitfa*, and *unc503* endogenous promoters, or, the lineages and/or temporal kinetic of transgene expression are not relevant to the disease.

In our zebrafish model, the relevant *PAX3-FOXO1* RMS cell type is supportive of CMV-mediated gene expression and requires a *tp53* mutation. These observations are substantiated by the mouse *Pax3-Foxo1* RMS model, in which deletion of *Tp53* or *Cdkn2a* is required for tumor development (*Keller et al., 2004*). In our zebrafish ARMS model, *tp53* mutations allow for *PAX3-FOXO1* + cells to survive during embryogenesis by inhibiting apoptosis resulting in later tumor formation (*Figure 2— figure supplements 1–2*). This may represent a unique necessity in animal models as both fusion-positive RMS vertebrate models of the disease require that *tp53* is mutant or lost. In zebrafish, beta-actin-driven *PAX3-FOXO1* is sufficient for tumorigenesis, providing a cleaner background for its functional assessment and greater conservation with the genetics of the human disease. One possible explanation is that zebrafish do not have p14 ARF orthologs, making it an unique model to understand the function of the fusion alone. Therefore, we pursued embryonic analyses of *PAX3-FOXO1* function using both the beta actin and CMV promoters due to their tumorigenic effects and unique genetic requirements. We found by performing analogous studies with BetaActin-*PAX3-FOXO1* and CMV-*PAX3FOXO1* that these expression constructs behaved comparably in an embryonic environment (*Figure 2*, *Figure 2—figure supplements 1–2*, *Figure 4*, *Figure 4—figure supplements 1–2*). Focusing on robust and conserved early oncogenic mechanisms that are consistent across multiple promoter models limited the possibility of artifactual observations.

By studying *PAX3-FOXO1* signaling in a complex developmental context, we identified a novel fusion-positive RMS biomarker, *her3/HES3*, that predicts more aggressive disease. *her3* is a transcription factor that contains a DNA-binding domain, an orange domain, and a tetrapeptide motif at the C terminus (*Hans et al., 2004*). In zebrafish and mouse models, *her3* is expressed in the neuroepithelial stem cells of the neural plate and inhibits their premature differentiation (*Hatakeyama et al., 2004*). The human ortholog of *her3*, *HES3*, is a member of the basic helix-loop-helix (bHLH) transcription factor family, which includes *MYOD1* and *MYOG*, that are known to be transcriptionally targeted by *PAX3-FOXO1* (14, 33). *HES3* acts as both a transcriptional repressor and activator, by indirectly or directly influencing transcription depending on its sub-cellular localization. *PAX3-FOXO1* regulation of *HES3* is likely indirect. Chip-seq data from rhabdomyosarcoma cell culture indicates that *PAX3-FOXO1* does not bind the *HES3* promoter; however, these results may differ in a dynamic developmental context (*Cao et al., 2010*). Copy number analysis of fusion-positive patient RMS tumors finds no amplification of *HES3*, indicating up-regulation is at the transcriptional level. Our data suggest that *PAX3-FOXO1* and *HES3* function linearly, with *HES3* being downstream of *PAX3-FOXO1*. The exact mechanisms of regulation will be addressed in future studies.

*HES3* contributes to more aggressive disease in patients when overexpressed in the context of the *PAX3-FOXO1* fusion, likely by a multi-variate platform of: (1) inhibiting terminal muscle differentiation, (2) modifying cellular plasticity to make cells that acquire the *PAX3-FOXO1* fusion more tolerant of its expression, and (3) inducing additional transcriptional changes. *PAX3-FOXO1* induction of *HES3* expression in a mesodermal lineage is ectopic, and *HES3* plays no known role in normal somitogenesis or myogenesis. We have found that mis-expression of *HES3* during embryonic development inhibits muscle differentiation. We note that *HES3* allows for *PAX3-FOXO1* cells to inappropriately persist during embryogenesis independently of inhibiting apoptosis. In human fusion-positive RMS, which typically lack *TP53* mutations (*Shern et al., 2014*), *HES3* overexpression could represent an alternative mechanism to facilitate *PAX3-FOXO1* + cell persistence and tumor initiation events. This role for *HES3* promoting *PAX3-FOXO1* + embryonic persistence is suggestive of *HES3*'s capacity in normal development to alter cellular plasticity. *PAX3-FOXO1* in combination with *HES3* could be challenging the epigenetic identity of mutated cells. Finally, we provide evidence that *HES3* may induce a more aggressive and/or metastatic program in *PAX3-FOXO1* + patient RMS, including the down-regulation of *MTSS1* and up-regulation of *MMP3* and *MMP9* (34, 35). Previous studies indicate that *MTSS1* is a target of the *PAX3-FOXO1* fusion (*Ebauer et al., 2007*); therefore, the modification of *MTSS1* expression, and additional genes integral in cell invasion could occur in part through *PAX3-FOXO1* induction of *HES3*. Most significantly, stratifying patient tumor data between *HES3* high and low expressing RMS elucidated translational molecular targets with immediate potential applications in the clinic such as co-expression of *FGFR4*, *ALK*, and *PARP1*.

Overall, this new *PAX3-FOXO1* zebrafish model of rhabdomyosarcoma identifies *her3*/*HES3* as a mediator of tumorigenesis. Further application of this model determined mechanisms of *PAX3-FOXO1*/*HES3* cooperation that result in more aggressive disease. Given the strengths of the zebrafish system, we can now apply it for drug screening and repurposing efforts. We envision the utility of our zebrafish *PAX3-FOXO1* RMS model being applied for transplant experiments, where generated tumors can be propagated and grafted into adult zebrafish and in developing zebrafish embryos for further study. Zebrafish embryos containing *PAX3-FOXO1* + tumor cells can be exposed to single or combinations of small molecules that effectively couple efficacy for tumor elimination with toxicity data in a vertebrate whole animal model. Additionally, zebrafish cancer models are a complementary approach to ongoing sequencing efforts of patient tumors. The functional significance of cooperating genetic mutations can be determined quickly and robustly. In our study, we find that human *PAX3-FOXO1* and *HES3* are active in zebrafish, and that biomarkers identified in this system are relevant in the human disease. We anticipate that additional application of this new *PAX3-FOXO1* driven tumor model will elucidate novel tumorigenic mechanisms and avenues for treatment.

# Materials and methods

**Key resources table**

| Reagent type (species) or resource | Designation | Source or reference | Identifiers | Additional information |
|---|---|---|---|---|
| Gene (*Homo sapiens*) | *HES3* | NA | NM_001024598 | |
| Gene (*Homo sapiens*) | *PAX3-FOXO1* | PMID:8275086; PMID:8221646 | NM_181457; NM_002015.3 | exons 1–7 of NM_181457 fused to exons 2–3 of NM_002015.3 |
| Gene (*Mus musculus*) | *Pax3* | NA | NM_008781 | |
| Gene (*Danio rerio*) | *her3* | NA | NM_131080 | |
| Strain, strain background (*Danio rerio* AB) | AB Wildtype | Zebrafish International Resource Center (ZIRC) | ZIRC:ZL1 | https://zebrafish.org/fish/lineAll.php |
| Strain, strain background (*Danio rerio* AB/TL) | AB\TL Wildtype | NA | | Cross of AB and TL |
| Strain, strain background (*Danio rerio* TL) | TL Wildtype | ZIRC | ZIRC:ZL86 | https://zebrafish.org/fish/lineAll.php |
| Strain, strain background (*Danio rerio* WIK) | WIK Wildtype | ZIRC | ZIRC:ZL84 | https://zebrafish.org/fish/lineAll.php |
| Strain, strain background (*Danio rerio* tp53$^{M214K}$ mutant) | tp53$^{M214K}$ mutant | PMID:15630097; available from ZIRC | ZIRC:ZL1057 | https://zebrafish.org/fish/lineAll.php |
| Genetic reagent | Tol2 transposase mRNA | PMID:16959904 | | Injected at 50 ng/uL |
| Cell line (*Mus musculus*) | C2C12 | ATCC | ATCC:CRL-1772 | Maintained in DMEM + 10% FBS+1X Antimycotic-Antibiotic |
| Cell line (*Homo sapiens*) | Rh30 | ATCC | ATCC:CRL-2061 | Maintained in RPMI-1640 + 10% FBS+1X Antimycotic-Antibiotic |
| Transfected construct (*Homo sapiens*) | CMV-HES3 (MYC and FLAG tagged) | Origene | Origene:RC224630 | |
| Transfected construct (NA) | CMV-Empty | this paper | | *HES3* excised using EcoR1 and Mlu1 and re-ligated |
| Antibody | MF20 (mouse monoclonal) | Developmental Studies Hybridoma Bank | DSHB:MF20c | 1:40 in cells, 1:100 in zebrafish embryo whole mounts, 1:1000 Western blot |

*Continued on next page*

Continued

| Reagent type (species) or resource | Designation | Source or reference | Identifiers | Additional information |
|---|---|---|---|---|
| Antibody | alpha-Tubulin DM1A (mouse monoclonal) | Cell Signaling | Cell Signaling:3873 | 1:1000 dilution |
| Antibody | MyoD 5.8A (mouse monoclonal) | Thermo | Thermo:MA5-12902 | 1:1000 dilution |
| Antibody | MYC 71D10 (rabbit monoclonal) | Cell Signaling | Cell Signaling:2278 | 1:1000 dilution |
| Antibody | GFP (rabbit polyclonal) | MBL International Corporation | MBL:598 | 1:1000 dilution |
| Antibody | GFP-488 (rabbit polyclonal) | Thermo | Thermo:A-21311 | 1:500 dilution |
| Antibody | Alexa 488 or 594 secondaries | Thermo | | 1:500 dilution |
| Antibody | HRP conjugate secondaries | BioRad | | 1:20000 dilution |
| Other | DAPI stain (ProLong Gold Antifade mounting media with DAPI) | Thermo | Thermo:P36931 | |
| Recombinant DNA reagent | p5E beta actin | PMID:17937395 | | |
| Recombinant DNA reagent | p5E cmv | PMID:17937395 | | |
| Recombinant DNA reagent | p5E mcs | PMID:17937395 | | |
| Recombinant DNA reagent | p5E ubi | PMID:21138979; available from Addgene | Addgene:27320 | |
| Recombinant DNA reagent | p5E unc503 | PMID:23444339; available from Addgene | Addgene:64020 | |
| Recombinant DNA reagent | p5E fli1a | PMID:17948311; available from Addgene | Addgene:31160 | |
| Recombinant DNA reagent | p5E mitfa | James Lister; available from Addgene | Addgene:81234 | |
| Recombinant DNA reagent | pmE beta globin splice acceptor | PMID:15239961 | | |
| Recombinant DNA reagent | pmE GFP2A | PMID:17941043 | | |
| Recombinant DNA reagent | pmE mCherry2A | PMID:17941043 | | |
| Recombinant DNA reagent | p3E *PAX3-FOXO1* | this paper | | attb2r/attb3 sites added with primers in *Supplementary file 3* by high-fidelity PCR |
| Recombinant DNA reagent | p3E *Pax3* | this paper | | attb2r/attb3 sites added with primers in *Supplementary file 3* by high-fidelity PCR |
| Recombinant DNA reagent | p3E *HES3* | this paper | | attb2r/attb3 sites added with primers in *Supplementary file 3* by high-fidelity PCR |
| Recombinant DNA reagent | p3E SV40 late poly A | PMID:17937395 | | |
| Recombinant DNA reagent | p3E 2A-mCherry | PMID:23462469; available from Addgene | Addgene:26031 | |

*Continued*

| Reagent type (species) or resource | Designation | Source or reference | Identifiers | Additional information |
|---|---|---|---|---|
| Recombinant DNA reagent | pDONRP2R-P3 (3' donor vector; attP2R-P3 flanking chlor/ccdB cassette) | Invitrogen | Invitrogen:pDONR P2R-P3 | Used to generate p3E's from this paper |
| Recombinant DNA reagent | pDestTol2pA2 destination vector | PMID:17937395 | | |
| Recombinant DNA reagent | BetaActin-GFP2A-pA | this paper | | Generated by Gateway Cloning |
| Recombinant DNA reagent | BetaActin-mCherry2A-pA | this paper | | Generated by Gateway Cloning |
| Recombinant DNA reagent | BetaActin-GFP2A-2AmCherry | this paper | | Generated by Gateway Cloning |
| Recombinant DNA reagent | BetaActin-mCherry2A-*Pax3* | this paper | | Generated by Gateway Cloning |
| Recombinant DNA reagent | BetaActin-GFP2A-*Pax3* | this paper | | Generated by Gateway Cloning |
| Recombinant DNA reagent | BetaActin-mCherry2A-*HES3* | this paper | | Generated by Gateway Cloning |
| Recombinant DNA reagent | BetaActin-GFP2A-*PAX3FOXO1* | this paper | | Generated by Gateway Cloning |
| Recombinant DNA reagent | CMV-GFP2A-pA | this paper | | Generated by Gateway Cloning |
| Recombinant DNA reagent | CMV-mCherry2A-pA | this paper | | Generated by Gateway Cloning |
| Recombinant DNA reagent | CMV-GFP2A-*Pax3* | this paper | | Generated by Gateway Cloning |
| Recombinant DNA reagent | CMV-mCherry2A-*HES3* | this paper | | Generated by Gateway Cloning |
| Recombinant DNA reagent | CMV-GFP2A-*PAX3FOXO1* | this paper | | Generated by Gateway Cloning |
| Recombinant DNA reagent | ubi-GFP2A-*PAX3FOXO1* | this paper | | Generated by Gateway Cloning |
| Recombinant DNA reagent | mitfa-GFP2A-*PAX3FOXO1* | this paper | | Generated by Gateway Cloning |
| Recombinant DNA reagent | fli1-GFP2A-*PAX3FOXO1* | this paper | | Generated by Gateway Cloning |
| Recombinant DNA reagent | unc503-GFP2A-*PAX3FOXO1* | this paper | | Generated by Gateway Cloning |
| Recombinant DNA reagent | SpliceAcceptor-GFP2A-*PAX3FOXO1* | this paper | | Generated by Gateway Cloning |
| Sequence-based reagent | GFP Morpholino | Gene Tools | Gene Tools: GFP Morpholino | (5' ACAGCTCCTCGCCC TTGCTCACCAT 3') |
| Commercial assay or kit | ApopTag Red In Situ Apoptosis Detection Kit | Millipore | Millipore:S7165 | |
| Commercial assay or kit | Affymetrix Zebrafish Gene 1.1 ST Array strip | Affymetrix | Affymetrix:901802 | |
| Commercial assay or kit (*Mus musculus*) | M384 Rhabdomy osarcoma 384 well panel | BioRad | BioRad:M384 Rhabdomyosarcoma | |
| Commercial assay or kit (*Homo sapiens*) | H384 Tumor Metastasis (SAB Target List) 384 well panel | BioRad | BioRad:H384 Tumor Metastasis (SAB Target List) | |
| Chemical compound, drug | Geneticin (G418) | Thermo | Thermo:10131027 | Select at 1 mg/mL |
| Software, algorithm | ImageJ | http://imageJ.nih.gov/ij | | |

*Continued on next page*

*Continued*

| Reagent type (species) or resource | Designation | Source or reference | Identifiers | Additional information |
|---|---|---|---|---|
| Software, algorithm | GraphPad Prism 7.0 c | https://www.graphpad.com | | |
| Software, algorithm | Rv3.3.1 | https://www.R-project.org | | |

## Zebrafish Husbandry

*Danio rerio* were maintained in an Aquaneering aquatics facility according to industry standards. Vertebrate animal work is accredited by AALAC and overseen by the UT Southwestern IACUC committee. AB, WIK, TL, and AB/TL were the wild-type lines used and were obtained from the Zebrafish International Resource Center (https://zebrafish.org). The *p53* mutant line, *tp53$^{M214K}$,* was a kind gift from Tom Look (*Berghmans et al., 2005*).

## Plasmids and cloning

Mouse *Pax3* coding sequence and human *PAX3-FOXO1* coding sequence was a gift from Steve Skapek. The *HES3* coding sequence with a MYC-DDK C-terminal tag was obtained from ORIGENE (RC224630). *Pax3*, *PAX3-FOXO1*, and *HES3* were cloned into the Gateway expression system (Thermo) by adding 5' and 3' ATT sites (attb2r/attb3) with primers and high-fidelity PCR (Thermo; *Supplementary file 3*). Purified PCR products were cloned into a 3'entry clone using the described protocol (*Kendall and Amatruda, 2016*). The tol2 kit beta actin promoter, cmv promoter, multiple cloning site, hsp70l promoter, 3' SV40 late poly A signal construct, and pDestTol2pA2 destination vector were used for construct generation and expression in zebrafish (*Kwan et al., 2007*). The ubi promoter was a kind gift from Len Zon (Addgene #27320) (*Mosimann et al., 2011*), the unc503 promoter from Peter Currie (Addgene #64020) (*Berger et al., 2013*), the fli1 promoter and 3' entry 2A-mCherry from Nathan Lawson (Addgene #31160 and #26031) (*Villefranc et al., 2007*; *Villefranc et al., 2013*), and the mitfa promoter from James Lister (Addgene #81234). Middle entry beta globin intron and splice acceptor was from Koichi Kawakami (*Kawakami et al., 2004*). The plasmids containing a GFP or mCherry viral 2A sequence were a gift from Steven Leach (*Provost et al., 2007*), and were sub-cloned into a middle entry Gateway expression system. Tol2 mRNA was synthesized from pCS2FA-transposase which was from Koichi Kawakami (*Urasaki et al., 2006*). The constructs generated and utilized in the described studies include: BetaActin-GFP2A-pA, BetaActin-mCherry2A-pA, BetaActin-GFP2A-2AmCherry, BetaActin-mCherry2A-*Pax3*, BetaActin-GFP2A-*Pax3*, BetaActin-mCherry2A-*HES3*, BetaActin-GFP2A-*PAX3FOXO1*, CMV-GFP2A-pA, CMV-mCherry2a-pA, CMV-GFP2A-*Pax3*, CMV-mCherry2A-*HES3*, CMV-GFP2A-*PAX3FOXO1*, ubi-GFP2A-*PAX3FOXO1*, mitfa-GFP2A-*PAX3FOXO1*, fli1-GFP2A-*PAX3FOXO1*, unc503-GFP2A-*PAX3FOXO1*, SpliceAcceptor-GFP2A-*PAX3FOXO1*.

## Zebrafish embryo injections

Zebrafish were injected at the single-cell stage with equimolar ratios of the described DNA constructs. Injection mixes contained 50 ng/µL Tol2 transposase mRNA, 25 or 50 ng/µL of BetaActin-GFP2A-*PAX3FOXO1* or CMV-GFP2A-*PAX3FOXO1*, and equivalent molar amounts of comparative plasmid DNA, 0.1% phenol red, and 0.3X Danieau's buffer. In knock-down experiments, 300 µM of GFP morpholino (5' ACAGCTCCTCGCCCTTGCTCACCAT 3'; Gene Tools), was added to the injection mixes in combination with the plasmid DNA.

## Zebrafish embryo survival and embryonic cellular persistence

For survival analysis of embryos, the total number of injected fish was counted, and then the resulting dead or alive embryos subsequently determined at 24, 48, and 72 hpf. Survival curves were plotted using GraphPad Prism 7.0 c (La Jolla, CA). For cellular persistence analysis during embryogenesis, embryos were injected at the single-cell stage with equimolar amounts of plasmid DNA and Tol2 transposase mRNA. At 24 hr post fertilization zebrafish embryos were dechorionated, and then individually imaged at 24 and 72 hr for GFP and mCherry expression using the exact same settings. In ImageJ, the positive pixel threshold was determined and then applied to quantify the number of GFP or mCherry-positive pixels for each embryo.

## Zebrafish embryo FACS sorting

For FACS sorting of GFP-positive zebrafish cells, zebrafish embryos were injected with purified DNA constructs and transposon mRNA at the single-cell stage and allowed to develop for 24 hr, after which they were deyolked and dissociated to single cells (*Manoli and Driever, 2012*). Using a MoFlo Cell Sorter (Beckman Coulter Life Sciences) live cells were gated, and then GFP + zebrafish cells were sorted out and collected in 1X Phosphate Buffered Saline (Thermo) on ice. Total RNA was immediately isolated using the RNeasy Microkit (Qiagen) and its integrity confirmed by Nanodrop on a spectrophotometer and a Bioanalyzer RNA Nanochip (Agilent). Total RNA was utilized for the Zebrafish Affymetrix Gene 1.1 Microarray strips which was run in-house.

## Zebrafish tumor collection, processing for RNA/DNA and histology, and tumor incidence

Zebrafish with tumors were humanely euthanized and screened under a Nikon SMZ25 fluorescent stereomicroscope to detect the fluorescent protein indicative of transgene expression. Fresh GFP + tumor tissue was resected and snap frozen in liquid nitrogen. Frozen tissue was subjected to DNA isolation with the DNeasy Kit (Qiagen) or total RNA isolation using the RNeasy Microkit (Qiagen). The remaining tumor specimen was placed in histology cassettes and fixed in 4% paraformaldehyde/1XPBS for 48 hr at 4°C (Fisher). They were then de-calcified in 0.5M EDTA for 5 days and mounted in paraffin blocks for microtome sectioning. Hematoxylin and eosin staining was performed on de-paraffinized slides. For tumor incidence curves, zebrafish were injected and then only those surviving past 30 days of age were included in the analysis. All zebrafish were screened under the fluorescent microscope to determine if they were GFP positive or negative. Zebrafish without GFP fluorescence were considered to be negative for transgene-dependent tumor formation. Zebrafish that were GFP positive were collected as described earlier, and the presence of malignancies confirmed by hematoxylin and eosin staining and visual review by a pathologist.

## Detection of apoptosis in whole-mount zebrafish embryos

Injected zebrafish embryos were fixed at 24 hr post fertilization in 4% paraformaldehyde/1XPBS for 24 hr at 4°C or for 2 hr at room temperature in scintillation vials. TUNEL staining was performed using the ApopTag Red In Situ Apoptosis Detection Kit (Millipore). A GFP counter-stain was performed with an anti-GFP antibody at 1:1000 (MBL International Corporation) and an Alexa Fluor 488 goat anti-rabbit IgG secondary (H + L) at 1:500 (Thermo), or an anti-GFP polyclonal antibody directly conjugated to Alexa Fluor 488 at 1:500 (Thermo). Images for rhodamine/GFP were taken using the same settings across embryos on a Nikon SMZ25 fluorescent stereomicroscope, and number of positive pixels in each embryo determined and analyzed in ImageJ.

## Microarray

Gene expression signatures of injected zebrafish embryos were compared for GFP controls, *Pax3*, or *PAX3-FOXO1* using the Affymetrix Zebrafish Gene 1.1 ST Array strip (Cat #901802). Each sample was run with technical triplicates. Microarray analyses were conducted using the R v3.3.1 environment (R core Team 2017). R: A language and environment for statistical computing. R Foundation for Statistical Computing, Vienna, Austria (https://www.R-project.org). Expression profiles were extracted and normalized using Robust Microarray Average (*Irizarry et al., 2003*). The intersection of statistically significant up-regulated genes for *Pax3* and *PAX3-FOXO1* conditions as compared to an injected GFP control was determined. Significance was assessed using a Welsh Two Sample t-tests. Genes unique to *PAX3-FOXO1* expression were eliminated if they did not contain a human ortholog (based on NBCI HomoloGene build 68 database), and genes were then rank ordered and prioritized based on fold change. Genes with statistically significant expression changes were analyzed using DAVID (https://david.ncifcrf.gov/) (*Huang et al., 2009a*; *Huang et al., 2009b*) to identify enriched Gene Ontology (GO) terms (http://www.geneontology.org/).

## RNA sequencing

Approximately 2 μg of total RNA was utilized for poly-A RNA enrichment and subsequent library preparation. Sequencing was performed on the Illumina NextSeq 500 Sequencing System with 2 × 75 bp paired end reads. For RNA-seq data, adapter removal and quality-filtering was conducted by

Cutadapt (*Martin, 2011*). Alignment to the reference zebrafish genome, build GRCz10, was performed by Bowtie2 (*Langmead and Salzberg, 2012*), an ultrafast and memory-efficient tool for aligning sequencing reads to long reference sequences. TopHat2 (*Kim et al., 2013*) was used for alignment. De novo assembly of reads into transcripts (including mRNA, lincRNA, and many other RNA species) and differential expression analysis were performed by Cufflinks and Cuffdiff (*Trapnell et al., 2012*). To detect the human *PAX3-FOXO1* fusion, we realigned all reads to the known junction sequence within *PAX3-FOXO1* fusion, which is shown in *Figure 1—figure supplement 2*. To reduce false positives, we required that (1) junction-spanning reads should contain no mismatch with known *PAX3-FOXO1* junction sequence and (2) junction-spanning reads with less than 6 bp matches on either gene was discarded, as suggested by an earlier study (*Li et al., 2011*).

## Immunofluorescence

In cells, the MF20 primary antibody was utilized for detection of myosin heavy chain at a dilution of 1:40 (Developmental Studies Hybridoma Bank; DSHB) in combination with an Alexa Fluor 488 goat anti-mouse IgG (H + L) secondary antibody at a dilution of 1:500 (Thermo). DNA was detected using ProLong Gold Antifade mounting media with DAPI (Thermo). MF20 staining in cells was performed as in *Kendall et al., 2012*. In zebrafish embryo whole mounts, the MF20 antibody was used for the detection of myosin at a concentration of 1:100 in combination with an anti-GFP polyclonal antibody directly conjugated to Alexa Fluor 488 at 1:500 (Thermo). The secondary antibody used was Alexa Fluor 594 goat anti-mouse IgG (H + L) at 1:500 (Thermo). Images were taken on a Keyence BZ-X700 fluorescent microscope.

## RNA extraction, cDNA synthesis, and qRT-PCR

Total RNA from zebrafish embryos, tumor tissues, and cells was isolated using the RNeasy Mini or Microkit (QIAGEN). cDNA was reverse transcribed from 200 ng-2μg of total RNA with the RT2 HT First Strand Synthesis kit (QIAGEN). qRT-PCR was performed on an ABI 7900HT using the SYBR-Green Master Mix (BioRad) and a 10 μL total volume in 384 well plates. See *Supplementary file 3* for primer sequences. Delta-delta Ct was calculated, and the calibrator plotted on the far left for every graph. Error bars indicate standard deviation and a student's t-test was performed on normalized Ct replicates to determine significance. The following BioRad qPCR arrays were utilized, M384 Rhabdomyosarcoma for mouse C2C12 cells and H384 Tumor Metastasis (SAB Target List) for Rh30 human RMS cells. Input cDNA was synthesized using the RT2 HT First Strand Synthesis kit (QIAGEN) including a synthetic mRNA as an internal quality control (BioRad). qRT-PCR arrays were run on the BioRad CFX384. Plotted is the fold change above and below the mean gene expression levels from all genes on the array.

## Protein extraction and western blots

Cells were harvested and lysed in radioimmunoprecipitation assay buffer (RIPA) with cOmplete Mini Protease Inhibitor Cocktail inhibitors (Sigma), and protein levels quantified using Qubit 3.0 Fluorometer (Thermo). Twenty micrograms of protein was denatured and loaded on a 4–20% gradient gel (BioRad), and then transferred to PVDF membranes at 4°C. Membranes were blocked in Casein + 0.1% Tween-20 (Thermo), and incubated overnight at 4°C with the following antibodies with agitation: MYOD monoclonal antibody 5.8A at 1:1000 (Thermo), MYC 71D10 monoclonal antibody at 1:1000 (Cell Signaling), alpha-Tubulin DM1A monoclonal antibody at 1:1000 (Cell Signaling), and MF20 at 1:1000 (Developmental Studies Hybridoma Bank). Goat anti-mouse IgG (H + L)-HRP conjugate and goat anti-rabbit IgG (H + L)-HRP conjugate secondaries were utilized at 1:20,000 (BioRad). Signal was detected using SuperSignal West Pico Chemiluminescent Substrate (Fisher). Membranes were imaged on the BioRad GelDoc XR + and quantification was performed in ImageJ.

## Cell culture and transfection

C2C12 and Rh30 were a kind gift from Steve Skapek. All cell lines were mycoplasma tested with the MycoAlert Mycoplasma Detection Kit (Lonza) within 6 months of their use and were negative. C2C12 was maintained in DMEM (Gibco) with 10% Fetal Bovine Serum (FBS; Sigma) and 1X Antibiotic-Antimycotic (Gibco) at 37°C in 5% $CO_2$. Rh30 were maintained in RPMI-1640 (ATCC) with 10% FBS (Sigma) and 1X Antibiotic-Antimycotic (Gibco) at 37°C in 5% $CO_2$. Differentiation media for C2C12s

included DMEM (Gibco) with 2% Horse Serum (Sigma) and 10 µg/mL of insulin (Fisher). Cells were transfected with Fugene HD Transfection reagent (Promega) of 3 µL:1 µg of DNA. Cells were selected for 30 days as sub-populations in 1 mg/mL of G418 (Thermo) prior to experiments and were maintained in 1 mg/mL of G418.

### Myogenic differentiation in cell culture

C2C12 cells were differentiated by plating 50,000 or 150,000 cells per well on 0.01% porcine/PBS-coated plates (24 wells or six welsl) in growth media + G418. If plating density varied from this it is indicated in the figure. For immunofluorescence C2C12 cells were plated on porcine coated glass coverslips. After 24 hours, cells were washed in 1XPBS, and differentiation media + G418 added. Cells were then fused for 6 days in differentiation media, with fresh media being added every other day. For fusion timepoints, cells were collected and pelleted after incubation in TrypLE Express Enzyme (Fisher), and frozen at −80°C for later total RNA isolation. For immunofluorescence, cells were fixed directly in the well with a 1:1 ratio of 4% paraformaldehyde:media at 37°C for 15 min. Subsequent washing and staining steps are detailed in *Kendall et al., 2012*.

### Cellular proliferation assays

Cells were seeded at 5,000 cells per well for C2C12 and Rh30 in a 24-well plate with four replicates per timepoint. Cells were maintained in GM supplemented with 1 mg/mL of G418 (Gibco) during the course of the experiment. At each timepoint, cells were fixed with 4% PFA for 15 min at room temperature, and stained with 0.0025% crystal violet (Alfa Aesar) in 20% methanol (Sigma) for 15 min. They were then rinsed and crystals were solubilized with 10% Acetic Acid (VWR). Absorbance was read on a plate reader at 595 nm.

### Soft agar colony formation assay

Colony formation assay strategy was adapted from *Borowicz et al. (2014)*. A bottom layer of 1.2% noble agar (Difco) in GM was plated with 1 mg/mL of G418 and allowed to solidify, followed by a top layer of 0.6% noble agar in GM with 1 mg/mL of G418 with 5000 cells per well in a six-well plate. Each well was maintained in 100 µL of GM + G418 that was changed three times per week. On day 30, wells were imaged for analysis, with three images being taken at 10X per well and then averaged. Six technical replicates were performed per sub-clonal population. Three biological replicates representing independent sub-clonal populations were included per experiment.

### Statistics

Statistical analysis for survival and tumor incidence curves, and Fisher's exact test, was performed using GraphPad Prism 7.0 c (La Jolla, CA). All other calculations were performed using two-tailed student's t test in Microsoft Excel Version 15.38. Sample sizes are provided in the figures or figure legends.

## Acknowledgements

This project is funded by a grant from Cancer Prevention and Research Institute of Texas (RP120685-P3) to JFA and the Ligue National Contre le Cancer (équipe labellisée). G Kendall is funded by a CPRIT postdoctoral fellowship through the UTSW Cancer Intervention and Prevention Discoveries training program, a QuadW-AACR Postdoctoral Fellowship for Clinical/Translational Sarcoma Research, a Young Investigator Grant from Alex's Lemonade Stand, and a Hartwell Foundation Postdoctoral Fellowship. S Watson was a recipient of a Ph.D. fellowship from Institut Curie. We thank the UTSW Molecular Pathology core and the UTSW Flow Cytometry core for exceptional services and for their expertise. We thank Tim Triche and Javed Khan for sharing RMS genomic data, and members of the Amatruda lab.

## Additional information

### Funding

| Funder | Grant reference number | Author |
| --- | --- | --- |
| Cancer Prevention and Research Institute of Texas | RP120685-P3 | James F Amatruda |
| American Association for Cancer Research | QuadW Foundation-AACR Fellowship for Clinical/ Translational Sarcoma Research | Genevieve C Kendall |
| Alex's Lemonade Stand Foundation for Childhood Cancer | Young Investigator Grant | Genevieve C Kendall |
| Hartwell Foundation | Postdoctoral Fellowship | Genevieve C Kendall |
| Ligue National Contre le Cancer | | Olivier Delattre |

The funders had no role in study design, data collection and interpretation, or the decision to submit the work for publication.

### Author contributions

Genevieve C Kendall, Conceptualization, Data curation, Formal analysis, Funding acquisition, Validation, Investigation, Methodology, Writing—original draft, Writing—review and editing; Sarah Watson, Collette A LaVigne, Whitney Murchison, Investigation, Writing—review and editing; Lin Xu, Data curation, Formal analysis, Investigation, Writing—review and editing; Dinesh Rakheja, Formal analysis, Writing—review and editing; Stephen X Skapek, Supervision, Writing—review and editing; Franck Tirode, Formal analysis, Supervision, Investigation, Writing—review and editing; Olivier Delattre, Formal analysis, Supervision, Writing—review and editing; James F Amatruda, Conceptualization, Data curation, Formal analysis, Supervision, Funding acquisition, Validation, Methodology, Writing—original draft, Project administration, Writing—review and editing

### Author ORCIDs

Genevieve C Kendall (iD) http://orcid.org/0000-0003-3775-2006
Dinesh Rakheja (iD) http://orcid.org/0000-0001-6888-7902
Franck Tirode (iD) http://orcid.org/0000-0003-4731-7817
James F Amatruda (iD) http://orcid.org/0000-0002-9901-2137

### Ethics

Human subjects: This study analyzed RNASeq and HuEx data from deidentified patient rhabdomyosarcoma samples. Samples were deidentified and histologic diagnosis and clinical information were compiled. All studies involving human subjects were conducted under a protocol approved by the UT Southwestern Institutional Review Board (approval # STU 102011-034).

Animal experimentation: UT Southwestern is a registered research facility with the US Department of Agriculture and is committed to comply with the Guide for the Care and Use of Laboratory Animals and all applicable federal, state and local regulations. These pertain to the purchase, transportation, housing and research use of animals. In addition, UT Southwestern is an AAALAC accredited institution that has developed institutional standards for the humane care and use of animals, which are maintained through published policies. Zebrafish research described in this study has been approved and conducted under the oversight of the UT Southwestern Institutional Animal Care and Use Committee (approval # 2016-101805). UT Southwestern uses the "Guide for the Care and Use of Laboratory Animals" when establishing animal research standards.

### Decision letter and Author response

Decision letter https://doi.org/10.7554/eLife.33800.031
Author response https://doi.org/10.7554/eLife.33800.032

# Additional files

## Supplementary files

• Supplementary file 1. Total number of zebrafish analyzed and resulting tumors for Beta Actin, CMV, and ubiquitin driven *PAX3-FOXO1* in wildtype AB or AB/TL strains or *tp53*[M214K/M214K] homozygous mutants.

DOI: https://doi.org/10.7554/eLife.33800.018

• Supplementary file 2. Total number of zebrafish analyzed and the mean age of screening for mitfa, fli1, unc503, and mcs (splice acceptor) driven *PAX3-FOXO1*.

DOI: https://doi.org/10.7554/eLife.33800.019

• Supplementary file 3. List of oligonucleotide primer sequences.

DOI: https://doi.org/10.7554/eLife.33800.020

• Transparent reporting form

DOI: https://doi.org/10.7554/eLife.33800.021

## Major datasets

The following dataset was generated:

| Author(s) | Year | Dataset title | Dataset URL | Database, license, and accessibility information |
|---|---|---|---|---|
| Triche T, Skapek S | 2018 | Exon level expression analysis from rhabdomyosarcoma patient tumors. | https://www.ncbi.nlm.nih.gov/geo/query/acc.cgi?acc=GSE114621 | Publicly available at the NCBI Gene Expression Omnibus (accession no: GSE11462) |

The following previously published datasets were used:

| Author(s) | Year | Dataset title | Dataset URL | Database, license, and accessibility information |
|---|---|---|---|---|
| Chen X, Stewart E, Shelat A, Qu C, Bahrami A, Hatley M, Wu G, Bradley C | 2013 | Data files for SJRHB RNA-Seq | https://www.ebi.ac.uk/ega/studies/EGAS00001000256 | Publicly available at the European Genome-phenome Archive (accession no. EGAD00001000878) |
| Shern JF, Chen L, Chmielecki J, Wei JS, Patidar R, Rosenberg M | 2014 | Genomic sequencing of Pediatric Rhabdomyosarcoma | https://www.ncbi.nlm.nih.gov/projects/gap/cgi-bin/study.cgi?study_id=phs000720.v2.p1 | phs000720.v2.p1; Genomic datasets from analysis of human subjects are dbGaP controlled-access data, available to users submitting an approved data access request through the dbGaP Authorized Access System (https://dbgap.ncbi.nlm.nih.gov/aa/wga.cgi?page=login) |
| Xu L, Zheng Y, Liu J, Rakheja D, Singleterry S, Laetsch TW, Shern JF | 2017 | Genomic sequencing of Pediatric Rhabdomyosarcoma | https://www.ncbi.nlm.nih.gov/projects/gap/cgi-bin/study.cgi?study_id=phs000720.v2.p1 | phs000720.v2.p1; Genomic datasets from analysis of human subjects are dbGaP controlled-access data, available to users submitting an approved data access request through the dbGaP Authorized Access System (https://dbgap.ncbi.nlm.nih.gov/aa/wga. |

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
