## [Decision Letter]

Thank you for submitting your article "*PAX3-FOXO1* transgenic zebrafish models identify *HES3* as a mediator of rhabdomyosarcoma tumorigenesis" for consideration by *eLife*. Your article has been favorably evaluated by Marianne Bronner (Senior Editor) and three reviewers, one of whom is a member of our Board of Reviewing Editors. The reviewers have opted to remain anonymous.

The reviewers have discussed the reviews with one another and the Reviewing Editor has drafted this decision to help you prepare a revised submission.

Summary:

Kendall et al. present data showing the development of a new RMS model in zebrafish. overexpress the human *PAX3-FOXO1* transgene found in alveolar rhabdomyosarcoma under a series of tissue specific and ubiquitous promoters to generate germline transgenic zebrafish that differentially develop tumors at different time points. Transient β-actin driven constructs expressing either *PAX3-FOXO1* or *PAX3* are employed to study the impact on zebrafish embryonic development and individual cell viability. *her3 (HES3* homolog) is identified as a differentially upregulated gene in *PAX3-FOXO1* zebrafish cells. Overexpression of *PAX3-FOXO1* or human *HES3* impairs mature muscle gene expression in zebrafish and overexpression of both increases cell survival. *HES3* overexpression also impairs mammalian muscle development, accelerates human rhabdomyosarcoma cell line proliferation and is found in patient tumors associated with poorer survival. Overall, this is an interesting and timely study that will add to our understanding of this disease. There are several issues raised by the three reviewers that would significantly strengthen the work:

Essential revisions:

1) The role of p53 and apoptosis in the observed phenotypesa) If *PAX3-FOXO1* results in increased cellular apoptosis (Figure 2E and I) then how do these cells survive to contribute to tumorigenesis? Is this apoptosis p53 dependent, thus the tumors develop only in the context of mutant p53? The authors do not directly address this and should examine apoptosis (e.g. TUNEL staining) in the context of mutant p53. *her3/HES3* may also be contributing as illustrated in Figure 4. Moreover, *PAX3* injected fish had reduced fluorescence compared with GFP or mCherry controls (32% vs. 70%) – is *PAX3* itself contributing to apoptosis? TUNEL staining or other bona fide apoptosis assays should be included to assess if *her3/HES3* or *PAX3* itself is specifically contributing to cell survival through an anti-apoptotic mechanism.

b) In Figure 2, the survival curves and representative embryos are convincing for the effects of the translocation. Have you examined the effect of adding in the p53 mutant in terms of survival from these transgenes? What is the onset of the RMS tumors driven by the CMV promoter in the *tp53* mutant background? Tumour presentation of 1.6-19 months of age is a huge range. This should be addressed – is it due to the different promoters or other factors? Were any fish followed beyond 19 months? If so, did any additional tumors develop? At a minimum, the authors should provide a table that summarizes each transgenic, type of tumor and onset.

2) The role of *HES3*a) Have you been able to identify any cell lines that have the *PAX3-FOXO1* translocation along with very high levels of *HES3*, and then tried to knock *HES3* down in that context? Does that affect tumorigenic behaviors? This would help give insight into how central *HES3* is, or whether it cooperates with the other genes you identified in Figure 3D (which seems likely). If such cell lines do not exist, could you speculate why *HES3* expression might be disfavored in the development of RMS cell lines?

b) As the authors point out, *HES3* is likely a transcriptional repressor, which would suggest that it is likely turning off a number of downstream pathways. From their original microarray studies in Figure 3, have they performed any type of analysis to indicate what signaling pathways might represent potential therapeutic targets in this situation? Even though therapeutic targeting is beyond the scope of this paper, it would be useful for the field to know what things they might go after in *HES3*+ tumors, and these pathways would be useful to bring up in the Discussion.

3) The effect of *PAX3-FOXO1* on muscle biology and embryo developmenta) *PAX3-FOXO1* is shown to be more toxic to developing embryos than *PAX3* alone (Figure 2H). For Figure 3 microarrays on whole embryos, could the gene expression differences be more indicative of cell populations lost in *PAX3-FOXO1*-expressing embryos? Do we know that the actual cells overexpressing either of these constructs are being assessed for alterations in *HES3* expression (i.e. is this a cell autonomous effect?).

b) Given that *PAX3-FOXO1* makes cells "sick" (Figure 2) and the fusion exerts some of its effects through *HES3*, could the decreased ability of myofibers to fuse with *HES3* overexpression in Figure 5 be nonspecific (i.e. are the cells unable to engage a specific differentiation program, or is this effect found with mis-expression of other, perhaps many, transcription factors?)

c) Along the same lines, while qPCR data suggests that *PAX3-FOXO1* zebrafish do not have muscle maturation by demonstrating low myoD expression levels, this is difficult to interpret in the context of mosaic animals and post-translational modifications of myoD. Thus, the zebrafish qPCR data should be supplemented with myoD immunofluorescence studies. Similarly, studies of myoD at the protein level (e.g. myoD and phospho-myoD by Western blot) should be done in the murine cell lines.

---

## [Author Response]

Essential revisions:1) The role of p53 and apoptosis in the observed phenotypesa) If PAX3-FOXO1 results in increased cellular apoptosis (Figure 2E and I) then how do these cells survive to contribute to tumorigenesis? Is this apoptosis p53 dependent, thus the tumors develop only in the context of mutant p53? The authors do not directly address this and should examine apoptosis (e.g. TUNEL staining) in the context of mutant p53. her3/HES3 may also be contributing as illustrated in Figure 4. Moreover, PAX3 injected fish had reduced fluorescence compared with GFP or mCherry controls (32% vs. 70%) – is PAX3 itself contributing to apoptosis? TUNEL staining or other bona fide apoptosis assays should be included to assess if her3/HES3 or PAX3 itself is specifically contributing to cell survival through an anti-apoptotic mechanism.

These questions are addressed in the experiments presented in new Figure 2—figure supplements 1-2. *PAX3-FOXO1* induces significant apoptosis in a wildtype zebrafish genetic background with both the BetaActin-*PAX3FOXO1* and CMV-*PAX3FOXO1* expression constructs. However, in the BetaActin-*PAX3FOXO1* model a sufficient number of cells escape to generate tumors (Figure 1B, F). In the new Figure 2—figure supplement 1, we explored whether this was true for the CMV-*PAX3FOXO1* model which differs from BetaActin-*PAX3FOXO1* in that it requires a *p53* mutation to generate RMS. We found that when injecting mosaic CMV-*PAX3FOXO1* into wildtype or *p53* mutant zebrafish that *PAX3-FOXO1*+ cells were more abundant in the *p53* mutant background. This was not true for other injected groups including the GFP controls and GFP-*PAX3*.

We next performed TUNEL to test whether this abundance of *PAX3-FOXO1*+ cells was due to a dysregulation of *p53*-mediated apoptosis. We found that in control GFP injected groups and *PAX3* injected groups there was no significant difference in the number of cells undergoing apoptosis when comparing wildtype and *p53* mutant zebrafish. However, there was a trend of increased apoptosis in the context of *PAX3* injections in wildtype zebrafish versus *p53* mutant zebrafish. These experiments underscored that *PAX3* is contributing in-part to an increase in embryonic apoptosis, but that the observed effect does not account for the entire spectrum of apoptosis that is induced by the *PAX3-FOXO1* fusion. In *PAX3-FOXO1* injected groups there was a significant decrease in apoptosis in *p53* mutant zebrafish as compared to their wildtype counterparts. Therefore, in the CMV-*PAX3FOXO1* model, the *p53* mutation is a sensitizer that allows these cells to survive and thus generate tumors later on. This model is outlined in Figure 2—figure supplement 2. This may represent a unique necessity in animal models; as both fusion-positive RMS vertebrate models of the disease (zebrafish presented here, and the *Pax3-Foxo1* mouse model), require that *p53* is mutant or lost for the generation of disease.

We detailed in Figure 4—figure supplement 1 how overexpression of human *HES3* contributed to the apoptosis phenotype in our CMV-*PAX3FOXO1* injected zebrafish embryonic model. We co-injected zebrafish with the following experimental design: 1) *HES3*+control, 2) *PAX3-FOXO1*+control, and 3) *PAX3-FOXO1*+*HES3*. Consistent with our previous data in the BetaActin-*PAX3FOXO1* model (Figure 4D-F) *HES3* overexpression allowed for GFP-*PAX3FOXO1*+ cells to persist. This trend was not true for GFP controls, *HES3* alone, or *PAX3-FOXO1* injected alone. We then performed TUNEL assays to determine if apoptotic inhibition was accounting for this increase in *PAX3-FOXO1*+ cells. This is not the case as *PAX3FOXO1*+*HES3* does not ameliorate the apoptosis phenotype. This model is detailed in Figure 4—figure supplement 2. However, these experiments did confirm that *PAX3-FOXO1* significantly increases apoptosis in the context of the CMV promoter, and that *HES3* alone does not.

b) In Figure 2, the survival curves and representative embryos are convincing for the effects of the translocation. Have you examined the effect of adding in the p53 mutant in terms of survival from these transgenes? What is the onset of the RMS tumors driven by the CMV promoter in the tp53 mutant background? Tumour presentation of 1.6-19 months of age is a huge range. This should be addressed – is it due to the different promoters or other factors? Were any fish followed beyond 19 months? If so, did any additional tumors develop? At a minimum, the authors should provide a table that summarizes each transgenic, type of tumor and onset.

We have performed the proposed survival experiment in Figure 2—figure supplement 1B. We found that injected CMV-*PAX3FOXO1* significantly decreased survival over three days of life in *p53* mutant embryos as compared to a wildtype background. CMV-*PAX3FOXO1* survival in a *p53* mutant background was also significantly different from *PAX3* injected into a *p53* or wildtype background, indicating that the observed activity is due in large part to the oncogenic fusion. These data suggest that *p53* mutants are unable to mount the appropriate response to an oncogenic insult, thus facilitating a more amenable environment for *PAX3-FOXO1*+ cellular survival and subsequent tumor development in those embryos that do survive.

The tumor incidence curve for RMS tumors in the *p53* mutant background is presented in Figure 1G. The tumor spectrum, onset, and details for all presented promoters and collected zebrafish tumors are presented in Supplementary files 1 and 2. We agree that the tumor spectrum and onset vary between the BetaActin-*PAX3FOXO1* and CMV-*PAX3FOXO1* models and attribute this discrepancy to being a mosaic model in which *PAX3-FOXO1* is under the control of different promoters in different genetic contexts. Every mosaic injection encompasses a different cellular context in which *PAX3-FOXO1* is being exposed to developmental cues. We have revised the text accordingly.

2) The role of HES3a) Have you been able to identify any cell lines that have the PAX3-FOXO1 translocation along with very high levels of HES3, and then tried to knock HES3 down in that context? Does that affect tumorigenic behaviors? This would help give insight into how central HES3 is, or whether it cooperates with the other genes you identified in Figure 3D (which seems likely). If such cell lines do not exist, could you speculate why HES3 expression might be disfavored in the development of RMS cell lines?

We agree that such experiments would delineate how essential *HES3* is to RMS cell survival. To identify HES3 expressing cell lines we have surveyed a panel of fusion-positive and fusion-negative RMS cell lines for *HES3* mRNA expression and HES3 protein expression. *HES3* mRNA is expressed at low levels and *HES3* protein is undetectable (data not shown). During normal development, *HES3* inhibits differentiation of neural stem cell populations. We discovered *her3/HES3’s* importance in the disease in a dynamic vertebrate developmental model system and have shown that *HES3* marks a less differentiated cellular population. We suspect *HES3* is lost during cell culture passaging over time due the artificial nature of culture conditions and the lack of developmental cues. Therefore, we will be exploring alternative models for studying *HES3* in culture systems including 3D cultures or primary patient xenografts. We anticipate 3D cultures and patient xenografts will more faithfully recapitulate *HES3*’s role in the development of RMS, will be a useful tool to identify novel markers of the disease, and will assess how stringently these models recapitulate each other and the patient.

b) As the authors point out, HES3 is likely a transcriptional repressor, which would suggest that it is likely turning off a number of downstream pathways. From their original microarray studies in Figure 3, have they performed any type of analysis to indicate what signaling pathways might represent potential therapeutic targets in this situation? Even though therapeutic targeting is beyond the scope of this paper, it would be useful for the field to know what things they might go after in HES3+ tumors, and these pathways would be useful to bring up in the Discussion.

We thank the reviewers for this suggestion. To address potential druggable targets we have analyzed RMS primary patient tumors and identified genes that are up and down-regulated in the context of *HES3* overexpression (Figure 7E). This analysis has identified a subset of druggable molecular targets with therapeutic implications; namely, the identification of *FGFR4, PARP1*, and *ALK* as being significantly overexpressed in *HES3*+ RMS. There are also targets that are downregulated which may be supplemented such as *DNAJC6*. We anticipate future studies will define the mechanistic contributions of these targetable genes as therapeutic strategies.

3) The effect of PAX3-FOXO1 on muscle biology and embryo developmenta) PAX3-FOXO1 is shown to be more toxic to developing embryos than PAX3 alone (Figure 2H). For Figure 3 microarrays on whole embryos, could the gene expression differences be more indicative of cell populations lost in PAX3-FOXO1-expressing embryos? Do we know that the actual cells overexpressing either of these constructs are being assessed for alterations in HES3 expression (i.e. is this a cell autonomous effect?).

In Figure 3 the microarray was performed on RNA isolated from FACS sorted GFP positive cells from injected zebrafish embryos. Therefore, we are confident the observed expression changes are cell-autonomous events and that *PAX3-FOXO1* is upregulating *her3* expression within the same cells.

b) Given that PAX3-FOXO1 makes cells "sick" (Figure 2) and the fusion exerts some of its effects through HES3, could the decreased ability of myofibers to fuse with HES3 overexpression in Figure 5 be nonspecific (i.e. are the cells unable to engage a specific differentiation program, or is this effect found with mis-expression of other, perhaps many, transcription factors?)

*HES3* is well tolerated in our overexpression systems including zebrafish models and stable cell lines. This indicates that *HES3* exerts a specific effect when it comes to dysregulating the engagement of myogenic differentiation. C2C12 or other myoblast fusion assays are established assays to determine genes that interfere with the fusion process. Such assays can distinguish between transcription factors or co-factors that inhibit or promote fusion. This has been the basis for high-throughput screens to identify specifically transcription factors that act as modifiers of the differentiation process (1). This strategy identified a subset of transcription factors that enhanced myogenic fusion, inhibited myogenic fusion, and a number that had no effect suggesting a specificity to the results. We have complemented our C2C12 mouse myoblast data with zebrafish developmental data in which *HES3* overexpression inhibits myogenesis in vivo (Figure 3C). We believe that *HES3* does not make the embryos sick, but rather likely promotes a modified differentiation state, and that this differentiation state is more capable of tolerating the oncogenic insult of the *PAX3-FOXO1*+ fusion. Therefore, since C2C12 myoblasts are healthy with *HES3* overexpression, and are capable of differentiating in some capacity after fusion induction, our interpretation is that *HES3* overexpression is engaging and inhibiting the differentiation program. The mechanism for this inhibition is likely indicative of *HES3*’s in vivo activity in combination with *PAX3-FOXO1*.

c) Along the same lines, while qPCR data suggests that PAX3-FOXO1 zebrafish do not have muscle maturation by demonstrating low myoD expression levels, this is difficult to interpret in the context of mosaic animals and post-translational modifications of myoD. Thus, the zebrafish qPCR data should be supplemented with myoD immunofluorescence studies. Similarly, studies of myoD at the protein level (e.g. myoD and phospho-myoD by Western blot) should be done in the murine cell lines.

In Figure 4C expression of *PAX3-FOXO1, HES3*, and *PAX3-FOXO1*+*HES3* had no significant effect on zebrafish *myod* or *myog* expression as compared to injected controls. We tested numerous antibodies that could potentially recognize zebrafish myoD, but no antibodies cross-reacted or faithfully recapitulated its known embryonic expression patterns. Therefore, in Figure 4B we have added a panel indicating that our mosaic injected transgenes do overlay with myosin protein, one of the muscle genes assayed in Figure 4C.

This data was supplemented with MyoD protein levels in C2C12 cells overexpressing HES3. We found that there was no difference in MyoD expression during the early initiation of differentiation; however, there was a trend towards decreased MyoD expression during terminal differentiation (Figure 5—figure supplements 1-2). Therefore, we concluded that in zebrafish and mammalian systems HES3 is not acting as a dominant negative on MyoD, but perhaps is inhibiting differentiation via engaging a different mechanism.

We have found that observations made from mosaic zebrafish are predictive of results from fluorescent cell populations that are sorted from embryos. For example, in Figure 3D GFP-*PAX3FOXO1*+ cells were FACS sorted from developing embryos, and *her3* exhibited a 21 fold up-regulation of expression. In Figure 3E we utilized zebrafish embryos injected with mosaic GFP-*PAX3FOXO1* and the entire embryo was implemented for RNA isolation and qRT-PCR of *her3* expression levels. In this context, *her3* exhibited ~5 fold increase in expression as compared to controls. Therefore, the mosaic expression system is sufficient to identify meaningful differences, but the effect might be muted in comparison to cell populations that are FACS sorted.

References:

1) Bakke J, Wright WC, Zamora AE, Ong SS, Wang YM, Hoyer JD, et al. Transcription factor ZNF148 is a negative regulator of human muscle differentiation. Sci Rep. 2017;7(1):8138.